

# Inversion of extensional basins parallel and oblique to their boundaries: Inferences from analogue models and field observations from the Dolomites Indenter, eastern Southern Alps

Anna-Katharina Sieberer[1], Ernst Willingshofer[2], Thomas Klotz[1], Hugo Ortner[1], Hannah Pomella[1]

[1]Department of Geology, University of Innsbruck, Innsbruck, 6020, Austria
[2]Department of Earth Sciences, Utrecht University, Utrecht, 3584 CB, Netherlands

*Correspondence to*: Anna-Katharina Sieberer (anna-katharina.sieberer@uibk.ac.at)

**Abstract.** Polyphase deformation of continental crust is analysed through physical analogue models for settings where platform-basin geometries at passive continental margins are subject to subsequent shortening and orogenesis. In a first stage, segmentation of the brittle and brittle-ductile models into basins and platforms is achieved by extension. Basins are partly filled with brittle material to allow for a strength differences between basin and platform realms, simulating relatively weaker, incompetent deposits of grabens surrounded by competent pre-rift basement or carbonate platform rock, respectively. In a second stage of deformation, contraction parallel to oblique (10 to 20 degrees) with respect to the basin axes has been applied leading to the inversion of earlier formed basins. The experiments show that the simple presence of an inherited platform-basin configuration controls the overall style of compressional deformation, no matter of including frictional or viscous basal décollements, of varying the rheology of the basin fill, or of changing platform-basin thickness ratios. Orientations of thrust faults change laterally across inherited platform-basin transitions throughout all experiments; higher obliquity of basin inversion leading to stronger curvature of thrusts with respect to the pre-existing rift axes. Variations in the strike of thrust fronts are accompanied by changes of the shortening direction along one single fault and time step. Furthermore, our models support localisation of deformation in areas of lateral strength contrasts, as platform-basin transitions represent. Reactivation of normal faults occurs in oblique basin inversion settings only, favourably at platform-basin transitions where the normal faults face the shortening direction. The amount and style of fault reactivation depend on the material used. Both parallel and oblique inversion experiments can be applied to polyphase deformed continental crust, as, e.g., the Dolomites Indenter of the eastern Southern Alps. Our models involving two phases of deformation, suggest that the whole tectonic evolution of the Dolomites Indenter is controlled by inherited features. Fault slip data and shortening directions from fold axes from our field case study along the western segment of the Belluno thrust of the Valsugana fault system support variations of thrust fault orientation and a lateral change in shortening direction (from SSW to SSE along strike) along one single fault. Based on our modelling results, we infer that this variability of shortening directions depends on inherited structures and do not necessarily reflect different deformation phases.

**Keywords.** Crustal-scale analogue modelling, parallel to oblique basin inversion, polyphase deformation, lateral strength contrasts, eastern Southern Alps, Dolomites Indenter.



## 1 Introduction

Shortening of passive continental margins is typically associated with the reactivation of inherited normal faults, inversion of sedimentary basins, and their incorporation in fold- and thrust belts (e.g., Turner and Williams, 2004; Cooper and Warren, 35 2020). Inverted sedimentary basins are known from numerous orogenic settings worldwide, from, e.g., the European Alps (Boutoux et al., 2014; Gillcrist et al., 2015; Granado et al., 2016; Oswald et al., 2018; Héja et al., 2022), the Apennines (Scisciani et al., 2001; Pace et al., 2014), the Pyrenees (Tavani et al., 2011; Mencos et al., 2015), Iberia (Ramos et al., 2017), the Atlas Mountains of Morocco (Beauchamp et al., 1999) and Algeria (Bracène and Froizon De Lamotte, 2002), and from the South American Andes (Kley and Monaldi, 2002; Giambiagi et al., 2003; Kley et al., 2005; Carrera et al., 2006). For 40 understanding complex and large-scale 3D tectonic patterns resulting from superposed extension and compression phases, tectonic inversion was already long studied through analogue (e.g., Buchanan and Mcclay, 1991; Sassi et al., 1993; Brun and Nalpas, 1996; Amiliba et al., 2005; Panien et al., 2005; Mattioni et al., 2007; Yagupsky et al., 2008; Cerca et al., 2010; Yamada and Mcclay, 2010; Bonini et al., 2012; Di Domenica et al., 2014; Granado et al., 2017; Deng et al., 2020; Zwaan et al., 2022) and numerical (Buiter et al., 2006; Panien et al., 2006; Buiter et al., 2009; Granado and Ruh, 2019; Ruh, 2019) modelling. 45 These studies confirm conceptual ideas (e.g., Sibson, 1985) and demonstrate that fault reactivation in inversion settings indeed strongly depends on the orientation and the dip angle of pre-existing discontinuities and on the rheology of rocks in foot- and hanging walls. Additionally, a combination of lateral stratigraphically controlled mechanical differences and pre-existing structures in basement or cover sequences often controls the location of so called transverse or transfer zones, which connect differing styles of deformation along strike of thrust belts (Thomas, 1990).

Within the European Alps, a prominent example of inverted sedimentary basins, which are now part of a fold- and thrust belt is the polyphase deformation history of the eastern Southern Alps. The relationship between Mesozoic extension, resulting in laterally distinct stratigraphic successions related to the structuration of the Adriatic crust into basins and platforms (Winterer and Bosellini, 1981; Sarti et al., 1992; Bertotti et al., 1993), and Cenozoic compression was intensively discussed by Doglioni (1987, 1991, 1992), as their interaction is crucial to understand the tectonic evolution of the eastern Southern Alps. Based on 55 field evidences from the Dolomites (eastern Southern Alps E of Bozen/Bolzano, W of the Cadore region and N of Valsugana fault system) and the Venetian Pre-Alps (eastern Southern Alps S of Valsugana fault system towards the Venetian plain) (Doglioni, 1991, 1992; Schönborn, 1999), and from the Friuli-Alps (eastern Southern Alps E of Cadore region, W of the Italian-Slovenian border) (Nussbaum, 2000), contrasting ideas for the evolution of inversion structures were proposed.

Through a series of crustal scale analogue models, we investigate the effect of an early extensional phase leading to 60 differentiation of the crust in platforms and basins on a later compressional phase, which is relevant for the discussion of inversion tectonics in settings such as the eastern Southern Alps. In particular, we test the hypothesis that pre-existing NNE-SSW trending normal faults are of paramount importance for understanding and explaining Paleogene to Neogene crustal deformation of the Dolomites Indenter. More precisely, we aim at demonstrating causal relationships between lateral changes of thrust fault orientations and the inherited fault-bound basin to platform transitions in extended crust. We substantiate our



findings by comparison with field observations, e.g., with the Belluno area, where the western (i.e., Trento) platform merges

into the eastern (i.e., Belluno) basin.

## 2 Geological setting of the Dolomites Indenter

In the evolution of the European Alps, the Adriatic plate is traditionally considered as rigid indenter (i.e., Adriatic indenter) (Schmid et al., 2004) and research mainly focused along its confining fault system, e.g., the Periadriatic fault system, the

Giudicarie belt, and the Valsugana and Montello fault systems (Fig. 1) and areas to the north elucidating collision and extrusion tectonics (Ratschbacher et al., 1991; Scharf et al., 2013; Favaro et al., 2017; Rosenberg et al., 2018). An indenter is known as a piece of rigid continental crust which, after collision, moves into weaker parts of an orogen (Tapponnier et al., 1986; Reiter et al., 2018). Following Schmid et al. (2004), we use the term Adriatic (micro)plate as part of greater Apulia which is located south of the Periadriatic fault system; Apulia being paleogeographically understood as consisting of all continental realms

between the Neotethys in the south and the Alpine Tethys in the north. However, the structure of the northernmost part of the Adriatic microplate within the eastern Southern Alps of Italy and Slovenia, referred to as Dolomites Indenter (Rosenberg et al., 2007), demonstrates significant internal deformation. This continental indenter contains the structural memory of Permian and Late Triassic to Early Jurassic extensional phases, which possibly controls thrust fault orientations related to Neogene to recent fold-and-thrust belt formation (Bosellini, 1965; Doglioni, 1992) in the eastern Southern Alps, with mainly in-sequence

deformation towards its external southern parts (Selli, 1998; Castellarin et al., 2006).

The eastern Southern Alps are bordered by the Pustertal-Gailtal fault (a part of the Periadriatic fault system) to the north, the Venetian plain to the south and the Giudicarie fault system and the Giudicarie belt to the west (Fig. 1a). The term eastern Southern Alps equals the term Dolomites Indenter, which in turn equals the eastern part of the Adriatic indenter. The Dolomites Indenter therefore represents the front of the Neogene to ongoing N(W)-directed continental indentation of Adria into Europe.

The indentation leads to the offset of the Periadriatic fault system along the Giudicarie fault system (Pomella et al., 2011), to the doming of the Tauern Window (Scharf et al., 2013; Schmid et al., 2013; Favaro et al., 2017; Rosenberg et al., 2018), to eastward lateral extrusion of crustal fragments of the Eastern Alps (Ratschbacher et al., 1991; Rosenberg et al., 2007), and to overall S-directed folding and thrusting of the eastern Southern Alps (Doglioni and Bosellini, 1987). In order to asses whether and how inherited platform-basin geometries affect younger, Alpine deformation, it is of importance to characterise the

Permian to Jurassic extension and Paleogene to Neogene shortening of the Dolomites Indenter.

[Figure 1]

### 2.1 Permian to Jurassic extensional phases

A first rifting event affecting the Adriatic crust started during the Lower Permian and was related to the opening of the

Neotethys (i.e., Meliata-Hallstatt Ocean) (Stampfli and Borel, 2002; Stampfli et al., 2002; Schmid et al., 2004). Extension lead



to the formation of N-S trending normal faults and ENE trending transfer faults (Doglioni, 1991), accompanied by the deposition of the up to ~2 km thick Athesian Volcanic Complex (Bosellini et al., 2007; Morelli et al., 2007; Marocchi et al., 2008; Brandner et al., 2016) (Figs. 1b, 2a). A second rifting event during the Late Triassic to Early Jurassic, associated with the opening of the Alpine Tethys (Sarti et al., 1992; Schönborn, 1999; Nussbaum, 2000; Masetti et al., 2012; Picotti and

Cobianchi, 2017) (or to the opening of the Neotethys Ocean according to, e.g., Vrabec et al. (2009)), segmented the Adriatic passive continental margin into submarine carbonate platforms and basins, which are bordered by N(NE)-S(SW) trending, crustal-scale normal faults (Winterer and Bosellini, 1981; Doglioni, 1991; Sarti et al., 1992; Bertotti et al., 1993; Selli, 1998; Busetti et al., 2010; Picotti and Cobianchi, 2017; Le Breton et al., 2021) (Figs. 1b, 2b). These normal faults reach into the upper and middle crust to depths of about 10 km (Martinelli et al., 2017) or 20 km (Masetti et al., 2012), and are associated

with minor normal faults reaching down to about 4 km depth (Pieri and Groppi, 1981).

From west to east, four major domains, bounded by N-S striking rift faults, were formed: the Lombardian basin, the Trento platform, the Belluno basin, and the Friuli platform (Winterer and Bosellini, 1981) (Figs 1b, 2). The Lombardian basin has the Ballino-Garda line, which is part of the Giudicarie belt, at its eastern border towards the Trento platform and therefore belongs to the western Southern Alps. The Trento platform is located east of the Ballino-Garda line and is split into a northern and a

southern part by the SW-NE trending Valsugana fault system. The northern Trento platform shows differential syn-sedimentary subsidence (Sarti et al., 1992; Martinelli et al., 2017) (Figs. 1b, 2) and approximately coincides with the extent of the Permian Athesian Volcanic Complex (Fig. 1b). The southern Trento platform (i.e., Venetian Pre-Alps) shows a thicker and more continuous sedimentary cover than the northern Trento platform (Fig. 2a). The Belluno basin is narrower in its W-E extent and shows a more complex geometry compared to basins west of the Ballino-Garda line as it is N-trending in its central

part, but NE-trending towards the Carnia region (Sarti et al., 1992). Whether and how the Belluno basin merges into the Slovenian basin, located north of the Friuli platform and NE of the Belluno basin (Fig. 1b), is topic of discussion (Smuc and Goričan, 2005; Van Gelder et al., 2015). The Friuli platform (i.e., Dinaric carbonate platform (Smuc, 2005)) shows stable shallow water sedimentation during most of the Jurassic and Cretaceous (Nussbaum, 2000; Merlini et al., 2002; Kastelic et al., 2008; Picotti and Cobianchi, 2017; Moulin and Benedetti, 2018) and merges into the vast Adriatic carbonate platform

(Vlahović et al., 2005) to the southeast.

**[Figure 2]**

## 2.2 Paleogene to Neogene shortening

Continental collision between Adria and Europe resulted in a first compressional phase (i.e., Pre-Adamello phase) within the western Southern Alps (i.e., Insubric Indenter west of the Giudicarie belt; Rosenberg et al. (2007)) which is mostly S-directed (Schönborn, 1990; Carminati et al., 1997), preceding the Adamello intrusion and therefore pre-middle Eocene in age (Zanchi et al., 2012; Zanchetta et al., 2013) and not documented within the eastern Southern Alps (Castellarin et al., 1992). During the



Paleogene, the eastern Southern Alps were in a foreland, pro-wedge position to Dinaric post-collisional shortening
(Ustaszewski et al., 2010). Especially the eastern part of the eastern Southern Alps, from Gadertal/Val Badia eastward, shows
mainly thin-skinned (Doglioni, 1987), WSW- to SW-directed Dinaric shortening (Doglioni and Bosellini, 1987; Caputo, 1996;
Keim and Stingl, 2000; Nussbaum, 2000) of about 10-15 km (Doglioni, 1992), whereas the western Venetian plain and the
western Dolomites were representing the foreland to the external Dinarides (Poli et al., 2021). After a short phase of extension
and transtension accompanied by volcanism (i.e., Veneto Volcanic Province, Zampieri (1995); Beccaluva et al. (2007)) within
the southern Trento platform NE of Verona (Fig. 1b), SSE-directed shortening in the eastern Southern Alps starts in Late
Oligocene (Fantoni and Franciosi, 2010; Vignaroli et al., 2020) to Miocene (Venzo, 1940; Castellarin and Cantelli, 2000)
times. From the Late Oligocene onwards, the eastern Southern Alps represent the retro-wedge of the Alpine orogen (Castellarin
and Cantelli, 2000). The Late Triassic to Jurassic platform-basin configuration was shortened and inverted; the Dinaric fold-
and-thrust belt got overprinted by overall SSE-directed deformation (Mellere et al., 2000; Placer et al., 2010). From north to
south, the major Neogene faults within the eastern Southern Alps are (i) the Valsugana fault system, including Valsugana and
Belluno thrusts and its N-directed back-thrusts, the Villnöss/Funes and Würzjoch/Passo delle Erbe faults (ii) the blind Bassano-
Valdobbiadene thrust with the Bassano anticline in its hanging wall and its N-directed back-thrust, the Val di Sella back-thrust
(Selli, 1998), and (iii) the blind Montello thrust with the Montello anticline in its hanging wall, representing the most external
structural feature of the eastern Southern Alps (Picotti et al., 2022).

The main shortening phase within the Dolomites Indenter (i.e., Valsugana phase) takes place during the Miocene (17-9 Ma),
according to the available but scarce thermochronological dataset (Zattin et al., 2006; Pomella et al., 2012; Heberer et al.,
2017). Crystalline basement in the hanging wall of the Valsugana thrust clearly shows Neogene thick-skinned thrusting in the
western part of the eastern Southern Alps (i.e., western Dolomites), but does not crop out in eastern part of the eastern Southern
Alps (Friuli region) above the main thrust. Serravallian to Tortonian sediments are deformed by the Valsugana thrust (Doglioni,
1992), representing the final push along the Valsugana fault system. Pliocene sediments are folded above the frontal triangle
zone of the Montello thrust (Doglioni, 1992; Ortner et al., 2016). According to recent studies of Anderlini et al. (2020), Jozi
Najafabadi et al. (2021), and Picotti et al. (2022), the Bassano-Valdobbiadene and the frontal Montello thrusts are seismically
active at present.

Mesozoic structures are frequently reactivated during the Neogene, as, e.g., the Permian normal Calisio and Schio-Vicenza
faults as strike-slip faults or cross-cut in shallower angles as, e.g., the dextral Paleo-Valsugana fault (Selli, 1998). Normal
faults related to platform-basin transitions are reactivated according to their dip angle and dip direction and lead to lateral
ramps for Neogene thrusts. According to Doglioni (1992), E-dipping Mesozoic normal faults have predominantly been cut
and involved in the alpine fold-and-thrust belt without major reactivations, whereas W-dipping Mesozoic normal faults seem
to often be strongly deformed and reused as thrust planes or by sinistral transpression, as it is the case for, e.g., the Giudicarie
belt. In the eastward prolongation of the Bassano-Valdobbiadene thrust, at the transition of Belluno basin and Friuli platform,
the Caneva line (i.e., Cansiglio line) represents a lateral ramp (Doglioni, 1991; Schönborn, 1999; Picotti et al., 2022). Neogene



structures stack platform regions onto basin regions, as, e.g., the Belluno thrust, which brings competent successions of the Trento platform onto more incompetent stratigraphy of the Belluno basin.

The amount of shortening along the Valsugana fault system is approximately 15 km (Selli, 1998). However, recent studies
show 6 to 8 km of shortening along the Belluno fault only (Zuccari et al., 2021), leading potentially to more shortening along the whole Valsugana fault system. According to Verwater et al. (2021), the amount of shortening across the eastern Southern Alps depends on competence contrasts and on thickness variations of sedimentary successions due to laterally heterogeneous paleogeographic domains. Therefore, competent platform successions show less shortening compared to basinal regions as, e.g., the Belluno basin, where the spacing of thrusts is especially narrow and the southern thrust front is located remarkable
far north (Doglioni, 1992).

## 2.3 Sedimentary succession and mechanical significance

In general, the stratigraphy of the eastern Southern Alps is characterised by a Variscian metamorphic basement, Permian volcanic rocks (Athesian Volcanic Complex, limited to the northern Trento platform), and a Late Permian to Neogene sedimentary succession (Winterer and Bosellini, 1981) (Fig. 2). The sedimentary succession shows minor differences in W-E
extent across Adria during the Upper Permian and Triassic, except stronger subsidence in the Lombardian basin (Sarti et al., 1992; Bertotti et al., 1993; Picotti et al., 1995) and stronger Ladinian volcanism in the western part of the eastern Southern Alps, but becomes strongly heterogeneous during Late Triassic to Jurassic extension (syn-rift sediments). Lateral and vertical facies changes known from several places within the eastern Southern Alps are, aside to normal faults, are other expressions of the platform-basin configuration of the Mesozoic passive continental margin (Abbots, 1989; Doglioni, 1992; Picotti and
Cobianchi, 1996; Selli, 1998; Franceschi et al., 2014; Picotti and Cobianchi, 2017).

Especially the northern Trento platform shows remarkable unconformities from the Jurassic onwards with, e.g., a fully eroded Mesozoic cover on top of the Athesian Volcanic Complex or a strongly reduced Jurassic (Winterer and Bosellini, 1981; Beccaro et al., 2002) to Cretaceous (Lukeneder, 2010) succession overlying the Athesian Volcanic Complex north of the Valsugana fault system (Fig. 2a). Within the southern Trento platform, the stratigraphic succession reaches into the Paleogene
to Miocene (Fig. 2a) with bioclastic to marly sediments (Doglioni and Carminati, 2008; Vignaroli et al., 2020). During the lower Jurassic, at the footwall of the rifted margins of the Trento and Friuli platforms, massive shallow water carbonates were deposited, whereas deep-water sediments occurred in the Belluno basin; oolitic limestone, shed from the platforms, transitions both realms (Masetti et al., 2012; Franceschi et al., 2014; Masetti et al., 2017; Picotti and Cobianchi, 2017) (Fig. 2b). The Slovenian basin shows a sediment thickness comparable to the Belluno basin (Fig. 2), but developed earlier (Ladinian) and is
deeper marine (Goričan et al., 2012; Rožič et al., 2018).

The main detachment horizon for thick-skinned deformation in the eastern Southern Alps is located at depth between 15 and 20 km, supported by recent local earthquake data (Jozi Najafabadi et al., 2021). For thin-skinned deformation, the two main detachment horizon are (i) evaporite-bearing facies associations of the Late Permian to Early Triassic Bellerophon Formation (Doglioni and Bosellini, 1987; Nussbaum, 2000), which thickens from W (no Bellerophon Formation west of Val




d'Adige/Etschtal) to E, reaching a maximum thickness of approximately 400 m in the E (Noé, 1987; Massari and Neri, 1997), and therefore the main detachment for SW-directed Dinaric structures and (ii) the alternate succession of evaporite, shales, and marls of the Carnian Raibl Group (Nussbaum, 2000) with a maximum thickness of about 250 m (De Zanche et al., 2000).

## 3 Analogue modelling approach

A series of 12 crustal-scale brittle and brittle-ductile analogue experiments provide insights in the structural evolution of
continental upper to middle crust subject to extension followed by contraction. The experiments are designed such to allow for the opening of multiple extensional basins separated by platform-type areas, a structural configuration that is frequently observed in passive margin settings where the continental shelf is dissected by graben structures and where the shelf transitions to the deeper basin (Mandl, 2000; Berra et al., 2009; Sapin et al., 2021). Subsequent contraction of the earlier formed graben structures of variable width allows for testing the influence of inherited structures and basin geometries on the style of a
younger contractional deformation phase, as well as timing and localisation of uplift of the inverted graben structures and their transition to platform areas. Key parameters of this study include the thickness differences between platforms and basins, the rheological stratification of the model crust, and the angle of obliquity between extensional and compressional deformation phases. The experiments have been designed to allow comparison with the Dolomites Indenter of the eastern Southern Alps, but are from a conceptional point of view also applicable to other regions affected by multiple deformation phases.

**3.1 Set-up and geometries of analogue models**

Model setups and the modelling results are described within a geographic frame where the north direction is aligned with the strike of the velocity discontinuity (VD) and thus the strike of the basin axes (Fig. 3a-c). All experiments are built on a table and on top of one fixed and two mobile plastic sheets of 1,0 mm thickness. Pre-deformation rotation of the fixed sheet by 10 and 20 degrees with respect to the extension direction allows for the formation of graben structures, which are at high angle to
the shortening phase (Fig. 3a-c). The mobile plastic sheets are attached to two separated engines which simulate extension by pulling the mobile sheets from underneath the fixed sheet at constant velocity but opposite directions (Fig. 3a-c). As such the transition from the fixed to the mobile plastic sheets predefines VDs along the western and eastern margins of the fixed plastic sheet. This kinematic boundary condition leads to asymmetric extension on either side of the fixed sheet comparable to proposed extensional geometries of the northern Adriatic plate margin (Sarti et al., 1992; Masetti et al., 2012). The extensional
phase was terminated after 5,0 cm of displacement of the eastern mobile sheet, whereas it continued to 9,0 cm of displacement at the western mobile sheet, producing sedimentary basins of different size comparable to the relatively narrow Belluno and wide Lombardian basins in the Southern Alps (Winterer and Bosellini, 1981; Bertotti et al., 1993; Picotti and Cobianchi, 2017).

**[Figure 3]**




Additional to pre-existing discontinuities and obliquity in basin inversion we apply lateral changes in mechanical stratigraphy to our experiments. Lateral strength contrasts result from (i) pre-existing deformation due to the first extensional phase, (ii) syn-extensional basin fill material, and (iii) lower thickness of the basinal succession compared to the platform succession. Regarding (ii), using different material for the basin fill than for platforms simulates relatively weaker, incompetent deposits

of grabens compared to surrounding competent pre-rift basement or carbonate platform rock, respectively. In our analogue models we tackle this by using, e.g., homogeneous layers of quartz sand for the initial, non-stretched model, simulating competent crust, which was extended in a first phase of deformation. Syn- to post-extensional sedimentation has been applied to the resulting grabens manually by sieving either quartz sand, feldspar sand or glass beads into the graben structures after approximately each cm of extension (Fig. 4a). The resulting grabens were filled up to a platform-basin thickness ratio of (i)

0,7 to 0,8 for underfilled basins using quartz sand only or (ii) 1,0 for filled basins using either quartz sand (model 4), glass beads (model 11) or feldspar sand (model 12), simulating relatively incompetent rock compared to the platforms (Fig. 4a). The thickness variation of underfilled basins (models 1-3 and 5-8) of (i) is a consequence of the handling technique. For (ii) we chose a platform-basin ratio of 1,0 for modelling the strength difference between platforms and basins only in terms of material properties, not in terms of varying material thickness.

Once the extensional phase is finished, the mobile plastic sheets are detached from their engines and fixed to the table. Subsequent contraction of the extensional basins is achieved by pushing the rigid backstop at constant velocity into the experiment. The velocity of the moving wall is 3,0 cm/h or 2,5 cm/h for brittle or brittle-ductile experiments, respectively. The compressional phase was terminated after 9,0 cm of shortening. Experiments consisting of brittle material only are confined by a rigid backstop in the north, aluminum bars in the east and west and have an open boundary in the south, whereas brittle

ductile models also have their southern boundary confined with aluminium bars to prevent the outflow of ductile material. A wider backstop was used in experiments 4 and 8 to model the evolution of the western basin in an oblique inversion setting with straight outlines only. All other oblique inversion models show a kink in the transition from the western basin to the western platform, because of limited space of setup arrangements, which is not of influence for the modelling results. An overview of modelling setups is provided in Table 1.


[Figure 4]

### 3.2 Model material and scaling

The analogue experiments presented in this study consist of brittle or a combination of brittle-ductile layers representing

continental upper crust, which is either entirely brittle or brittle with a ductile layer at the base (Fig. 4a). The latter simulates crustal layers below the brittle-ductile transition (models 6 and 8). Layers of colored dry quartz sand represent the brittle pre-rift crust in all models, whereas the ductile layer consists of polydimethylsiloxane (PDMS silicon polymer), mixed with Rhodorsil gomme. The ductile material has a density of 1500 kg/m³, a viscosity of 3,8 x 10⁴ Pas, and shows slightly non-



Newtonian behaviour (n = 1,15). The properties of the ductile material have been determined with a pycnometer and a coni-
cylindrical viscometer, respectively. Variations to these setups entail a layer of glass beads at the base of the brittle crust
simulating a weak, frictional décollement comparable to what has been used by Cotton and Koyi (2000). The influence of
sediment strength on subsequent deformation geometries during the inversion phase is accounted for by varying the granular
material representing the syn-extensional sediments. These include quartz sand (model 4), feldspar sand (model 12) and glass
microbeads (models 9 and 11). The mechanical properties of all brittle materials used in this study are summarized in Table 2
and are described in detail in Willingshofer et al. (2018) and Klinkmüller et al. (2016).

**[Table 1]**

**[Table 2]**

Scaling of all models follows standard scaling procedures described in the pioneering works of Hubbert (1937) and
Weijermaars and Schmeling (1987). With a length-scale ratio of L* (Lmodel/Lnature) = 1,25 x 10⁻⁶, 1,0 cm in the model
represents 8,0 km in nature (1:800.000). As such the initial and final model widths (after extension) scale to 352 and 464 km
in nature, respectively. For modelling contraction of oblique rift structures, models 4 and 8 have been increased in size to
minimize boundary effects; a larger backstop of 85 cm, scaling to 680 km in nature, was used. For the adopted length scaling
the brittle-ductile transition of our analogue models is at a depth of 16 km in nature, which agrees with earlier published
estimates of the brittle-ductile transition in the eastern Southern Alps (Willingshofer and Cloetingh, 2003; Viganò and Martin,
2007; Laubscher, 2010). At this depth the vertical stress (i.e., lithostatic stress) in, e.g., the platform regions of our experiments,
is 590 Pa, corresponding to 880 MPa in nature, which results in a stress-scale ratio of 6,7 x 10⁻⁷. Strength profiles (Fig. 4b) are
representative for the initial conditions and have been calculated following Brun (2002). The density-scale ratio ρ*
(ρmodel/ρnature) yields 0,54 as the density of our model material, e.g., quartz sand, is 1500 kg/m³, representing densities of 2800
kg/m³ of natural upper to middle crust (e.g., Ebbing, 2004; Faccenda et al., 2007; Šumanovac et al., 2009). The velocity for
our brittle-ductile models is scaled after a time-scale ratio t* = 1,93 x 10⁻¹¹ calculated as:

$$t^* = \frac{1}{\varepsilon^*} \qquad (1)$$

with ε* referring to the shear strain rate ratio equaling:

$$\varepsilon^* = \frac{v^*}{L^*} \qquad (2)$$

where the velocity ratio v* is 6,49 x 10⁴ and scales 2,5 cm/h of model velocity rate to 0,33 cm/a to nature, representing 50 km
of shortening within a period of 15 Ma. This shortening rate correlates well with estimates form orogenic belts such as the
eastern Southern Alps (Doglioni, 1987, 1991; Nussbaum, 2000). The velocity of the moving wall for brittle-ductile experiments
simulates a compressional model strain rate of 1,13 x 10⁻³, corresponding to 2,18 x 10⁻¹⁴ in nature. The viscosity of the ductile
model material of 3,8 x 10⁴ Pas scales to natural viscosities of approximately 10²¹ Pas, representing, according to, e.g., Hirth
et al. (2001), quartz dominated ductile middle curst below the brittle-ductile transition.



## 3.3 Monitoring and analysis of experiments

During the experimental runs, the model surface is monitored by top-view photographs (using Panasonic Lumix DC-G9 with 20,3 megapixels) and a 3D surface scanner in order to trace changes in surface deformation. Top-view pictures and 3D surface scans have been taken at regular intervals. For brittle and brittle-ductile experiments, the translation displacement between two successive top-view photographs is 0,15 cm and 0,125 cm, between two surface scans 0,75 cm and 0,625 cm, respectively. Adding coffee grains on the model surfaces facilitated calculating incremental particle displacements using particle image velocimetry (PIV) analysis (PIVlab, Thielicke and Stamhuis, 2004) and PIV-derived strain analysis (strainmap, Broerse et al., 2021). The strainmap of Broerse et al. (2021) further enables to determine strain types (e.g., extension, strike-slip, shortening and transitions) occurring during the respective model runs from the ratio of the largest to the smallest principal strain. The conversion of 3D surface scan data into digital elevation models enables tracing of the topographic evolution during the compressional deformation phase. After the model run, layers of black and natural coloured quartz sand are implemented as post-kinematic covers in order to preserve topography. Water sprinkled on the sand increases cohesion and enables sectioning the experiments for studying internal deformation in 2D. Photographs of 2D cross-sections (using Panasonic Lumix DC-G9 with 20,3 megapixels) are then put back together to a quasi 3D model using Midland Valley's MOVE software, in order to follow fault geometries laterally throughout the model.

## 3.4 Limitations and simplifications of analogue models

Potential influences leading to strain localisation like (i) surface processes (e.g., erosion, transport or deposition of sediments), (ii) isostasy or (iii) temperature dependence through depth are absent in physical analogue modelling techniques and hence mark major limitations. The viscosity of the ductile layer is therefore invariant to depth, what is widely accepted in analogue modelling studies (Davy and Cobbold, 1991), but can be adapted through implementing appropriate uniform viscosities. Regarding (ii), the absence of an asthenospheric layer excludes isostatic compensation of the model layers and therefore leads, in brittle-ductile models, to thickening of the ductile base layer. Ductile thickening accommodates deformation during the shortening phase mainly in uplift, localising the strain in systems of symmetric fore- and back-thrusts, which develop close to the moving wall. Despite those simplifications, the presented experiments resemble first-order deformation processes of a polyphase deformed continental crust and possible influences of erosion and sedimentation on the localisation of faults or on fault geometries in general (Graveleau et al., 2012) are acknowledged.

## 4 Modelling results

In the following sections, the results of our analogue modelling study are described, starting with a detailed description of the "reference model" (model 1, Fig. 5). Thereafter we comparatively describe the models based on interpreted top-view photographs, interpreted cross-sections of the final stage of the specific experiment, PIV analyses, PIV-derived strain analyses, and by the use of topographic profiles extracted from digital elevation models. Note that the transpressional deformation at the





western side of the western basin of model 4 and model 8 is a result of the modelling setup with a wider backstop and not relevant for comparison with the natural prototype. The term thrust system (i.e., thrust sheet) is used in the following for a
combination of fore-thrust and accompanying back-thrust(s).

## 4.1 Reference model, model 1

Model 1 (Fig. 5) is an entirely brittle model (i.e., quartz sand both for platforms and basin fillings) where basin inversion is parallel to the strike of the rift structures (Table 1, Fig. 3a).

During extension, two asymmetric grabens, separated by one platform, form, each with a major, high-angle normal fault above
the basal VDs (i.e., west and east of the western platform) (interpreted top-view picture in Fig. 5a, section c-c' in Fig. 5g). As the basal kinematic boundary condition is opposite for both graben structures, the major normal faults dip in opposite directions. Accompanying the major normal faults with large offset (e.g., 1,2 cm offset along major normal faults of model 1 in section c-c' in Fig. 5g) on the stationary (i.e., referring to the fixed plastic sheet beneath the deformable parts of the model) side of the evolving graben, this first extensional phase also creates a large number of normal faults with smaller offsets on the
moving (i.e., referring to the mobile plastic sheet beneath the deformable parts of the model) side. All normal faults evolve perpendicular to the direction of extension, are parallel to each other, appear straight, and dip towards the center of the main graben with an average dip angle of 63°. At the end of extension, model 1 consists of two basins and two platforms, which are perpendicular to the backstop (Fig. 5a, section c-c' in Fig. 5g). The crust was thinner at the locations of the basins (1,4 cm average thickness) compared to the platforms (2,0 cm), leading to a platform-basin thickness ratio of 0,7 representing lateral
mechanical strength differences (Fig. 4b, section c-c' in Fig. 5g), which, simulates overall weaker basin domains (e.g., alternation of limestone, marl, clay) compared to non-stretched platform areas (e.g., basement rock, carbonate platforms). Note that the strike-slip fault north of the eastern platform (Fig. 5a-d) is a result of the modelling setup, which bears no major implication for the modelling result during contraction.

Early stages of parallel basin inversion (Fig. 5b) already show curved thrust faults forming at platform-basin transitions (thrusts
1 of thrust sheet I in Fig. 5b), where the relatively weaker basins transitions to the stronger platforms. The evolving thrust faults clearly show anastomosing patterns at their fronts and cross-cut pre-existing normal faults without reactivation of the latter.

[Figure 5]


After 4,5 cm of shortening (Fig. 5c), a second, in-sequence thrust system II evolves, showing a curved thrust front, which is more distinct at the transition from the central western platform to the western basin than into the eastern basin. The cumulative strain map after 4,5 cm of shortening (Fig. 5d) shows shortening close to the backstop especially in and close to basinal areas (grey arrows in Fig. 5d), extension in the hanging wall of thrust system I within platform areas, and shortening at the fronts of
both thrust systems I and II. Parallel basin inversion is here most probably compensated by back-thrusting, as fault reactivation




in such a high angle is not the favoured mechanism. Cross-sections of the final shortening stage of model 1 (sections a-a' and b-b' in Fig. 5g) reveal the platform hosted back-thrusts 2, 4, 5 (section a-a' in Fig. 5g) and basin hosted back-thrusts 2, 3, 6 (section b-b' in Fig. 5g) which were active at 4,5 cm of shortening. The prominent back-thrust 6 of the eastern basin (section b-b' in Fig. 5g) is traceable on the model surface from 4,5 cm of shortening onwards (Fig. 5c-f).

By the end of shortening (Fig. 5e-f, sections a-a' and b-b' in Fig. 5g), the thrust front of the third and most external thrust system III is located further towards the foreland on platforms than in basinal areas (Fig. 5e-f, sections a-a' and b-b' in Fig. 5g). Pre-existing normal faults are cross-cut and transported piggy back as shown in Fig. 5e. The localisation of deformation in basinal areas is represented by backstepping of oblique thrust fronts towards the hinterland. The extensional cumulative strain type which dominates the hanging walls of thrust systems I and II in platform areas (Fig. 5f), is a result of the passive

uplift of thrust systems in the hinterland of the model orogen with in-sequence character. Passive uplift occurs along reactivated fore- and accompanying back-thrusts of the specific thrust systems. Fault reactivation is shown by two separate back-thrusts (back-thrusts 4 and 5 in section a-a' in Fig. 5g), which root in one single fore-thrust 3 (section a-a' in Fig. 5g). Both back-thrusts 4 and 5 cross-cut fore-thrust 1 and therefore indicate a younger age. Due to this reactivation of fore-thrust 3 and the cross-cutting of fore-thrust 1, thrust system I with its initial fore-thrust 1 and back-thrust 2 (section a-a' in Fig. 5g) steepens

along fore-thrust 1 and gets uplifted passively by younger thrusts 3 to 5 (section a-a' in Fig. 5g).

Overall, the style of thrusting documented in model 1 is a combination of foreland vergent pop-up systems, with in-sequence thrusting. Back-thrusts are either reactivated several times or formed new, meaning existing foreland-directed thrust faults being cross-cut by younger back-thrusts (e.g., fore-thrust 1 being cut by back-thrusts 4 and 5 in section a-a' in Fig. 5g). The style of thrusting varies laterally, across platforms and basins, as both ramps and flats are longer on platforms (e.g., 4 cm ramp

length of thrust 3 in section a-a' in Fig. 5g) compared to those of basinal areas (e.g., 1,4 cm ramp length of thrusts 5 and 7 in section b-b' in Fig. 5g), what is in accordance with the undulating surface expression of the thrust systems in map-view (Fig. 5e-f). The wavelengths of thrusts (i.e., the distance between the transition of lower flat to ramp of two adjacent thrusts) are longer on platforms compared to basin realms in parallel inversion models, e.g., 8,6 cm for platform hosted thrusts 3 to 6 (western platform, Fig. 5e-f, section a-a' in Fig. 5g) and 6,0 cm for basin hosted thrusts 7 to 8 (eastern basin, Fig. 5e-f, section

b-b' in Fig. 5g). As normal faults are cut rather than reactivated during shortening, the basins are inverted and uplifted by closer spaced thrust imbrication compared to platforms (e.g., thrusts 4 and 7 in section b-b' in Fig. 5g). This difference is related to thickness variations of the crust at the onset of shortening.

**4.2 Parametrical study**

In the following, we compare modelling results with variations in (i) obliquity of inversion (Fig. 6), (ii) décollement rheology

at base of the model (Fig. 8), and (iii) strength of the basin fill (Fig. 10) with respect to the reference model (model 1, Fig. 5).





### 4.2.1 Influence of obliquity of inversion

Models 2 (Fig. 6a-c) and 3 (Fig. 6d-f) represent models where the inversion has taken place at angles of 10 and 20 degrees with respect to the rift axes (Table 1, Fig. 3b-c), respectively. Similar to model 1, both models are brittle only, have a platform-basin thickness ratio of 0,7 to 0,8 (i.e., underfilled basin) (Table 1) and a syn-extensional basin fill consisting of quartz sand.

Overall, models 2 and 3 produced close to similar basin-platform geometries by the end of the extensional phase (compare Fig. 5 and Fig. 6).

The analysis of oblique basin inversion models shows that the curvature of the thrust fronts in map-view increases with obliquity (compare Fig. 6a-b for 10° obliquity with Fig. 6d-e for 20° obliquity). In models 2 and 3, this curvature is maximum at platform basin transitions where the strike of the trusts is at angles of 50 to 60 degrees with respect to the shortening direction

(e.g., thrusts 5 of model 2 and 3 shown in Fig. 6a-b and Fig. 6d-e, respectively). We note that angles of maximum 45° are obtained for models with parallel basin inversion.

Platform thrusts link with basin thrusts preferred at the eastern border of the grabens, e.g., at the transition from the eastern basin to the eastern platform (Fig. 6a-b, d-e), whereas in model 1, thrusts connect in the center of the respective basin (Fig. 5e-f). Therefore, the location of lateral ramps and transfer zones shifts in oblique basin inversion models further towards the

platform-basin transition. Additionally, the difference of wavelengths of thrusts between platform and basinal areas is more pronounced in experiments with oblique basin inversion (Fig. 6c, f) compared to models with parallel basin inversion (Fig. 5g). The wavelength of thrusts is, e.g., for platform hosted thrusts 5 to 7 of model 2 (western platform, Fig. 6a-b, section a-a' in Fig. 6c) is 12 cm, whereas it is only 0,8 cm for basin hosted thrusts 6 to 7 (eastern basin, section b-b' in Fig. 6c). Comparable differences in wavelength of thrusts from platforms to basins shows model 3, e.g., 7 cm for platform hosted thrusts 3 to 5

(western platform, Fig. 6d-e, section a-a' in Fig. 6f) and 1 cm for basin hosted thrusts 4 to 5 (eastern basin, section f-f' in Fig. 6f). Both in model 2 and model 3, ramps and especially flats of thrusts on platforms are particularly longer (e.g., 5 cm ramp length of thrust 5 of model 2 in section a-a' in Fig. 6c and 4 cm ramp length of thrust 5 of model 3 in section e-e' n Fig. 6f) compared to thrusts within basins (e.g., 4 cm ramp length of thrust 7 of model 2 in section b-b' in Fig. 6c and 3 cm ramp length of thrust 8 of model 3 in section f-f' in Fig. 6f).


**[Figure 6]**

Models with oblique basin inversion seem to favour the development of laterally shorter (thrust sheets III and IV in model 2 in Fig. 6a-b and thrust sheets III and IV in model 3 in Fig. 6d-e) and isolated thrust systems (thrust sheet II in model 2 in Fig.

6a-b and thrust sheet II in model 3 in Fig. 6d-e). Model 2 (Fig. 6a-c) and model 3 (Fig. 6d-f) both show an initial thrust system I, which was laterally continuous over the entire model width, comparable to parallel basin inversion (Fig. 5). However, thrust system II of models 2 and 3 is limited to the transition of the eastern basin to the western platform only. Thrust system III in model 2 (Fig. 6a-c) shows, although very narrow in N-S direction within the eastern basin (Fig. 6a-b, sections b-b' and c-c' in

Fig. 6c), a W-E extent over the entire width of the model, whereas thrust system IV in model 3 (Fig. 6d-f), with increased
obliquity compared to model 2, terminates within the eastern basin and does not reach the eastern platform. Thrust sheets IV
and V in model 2 (Fig. 6a-c) both terminate within the eastern basin, either as western or eastern termination. Thrust sheet III
in model 3 (Fig. 6d-f) is isolated and only present within the eastern platform and the eastern basin. This is in contrast with
parallel basin inversion, where thrust systems I to III can be found with a continuous development of the deformation front all
over the model's W-E extent (model 1, Fig. 5).

In the case of oblique basin inversion, parts of platform sequences get thrusted over basinal sequences as shown at the transition
of the western platform to the eastern basin along thrust 7 of model 2 (Fig. 6a-b) or thrust 5 of model 3 (Fig. 6d-e). Additionally,
parts of basinal sequences get thrust over platform sequences (thrust 4 in section c-c' in Fig. 6c, thrust 8 in section f-f' and
thrust 5 in section g-g' in Fig. 6f). Otherwise, thrust faults cross-cut pre-existing normal faults and either overthrust them
and/or transport them piggy back. In model 2, pre-existing normal faults are mostly cross-cut by younger thrust faults
throughout the experiment, with an exception within the imbricated stack of thrusts of the eastern basin (thrusts 4-7 in section
b-b' in Fig. 6c), where pre-existing normal faults get reactivated in oblique slip mode (thrusts 4, 6-7 in section b-b' and thrusts
5 and 7 in section c-c' in Fig. 6c). Thrusts 4, 6 and 7 of section b-b' (Fig. 6c) and thrusts 5 and 7 of section c-c' (Fig. 6c) of
model 2 (Fig. 6a-c) incorporate fully reactivated normal faults up to their initial termination at the boundary between syn- and
post-rift sediments (section d-d' in Fig. 6c) and cut through post-rift sediments to the model surface. Thrusts 6 and 7 of section
c-c' (Fig. 6c) are, together with non-reactivated normal faults, passively steepened by the younger in-sequence fore-thrust 8
(section c-c' in Fig. 6c). In model 3, oblique reactivation of pre-existing normal faults as thrust faults is more common further
in the hinterland of the model orogen (thrusts 5-6 in section f-f' and thrust 5 in section g-g' in Fig. 6f) compared to model 2
where fault reactivation is located further towards the foreland (thrusts 4, 6-7 in section b-b' and thrusts 6-7 in section c-c' in
Fig. 6c). In model 3, normal faults are (i) if reactivated, fully reactivated as compressional faults (e.g., thrusts 5-6, 8 in section
f-f' and thrusts 4 and 7 in section g-g' in Fig. 6f), comparable to those in model 2, (ii) cross-cut by younger back-thrusts (e.g.,
thrust 9 in section g-g' in Fig. 6f) and (iii) favoured areas of deformation re-localisation (e.g., thrust 7 in section g-g' in Fig.
6f grows towards the pre-existing normal fault and reactivates it).

Regarding the obliquity of slip along thrust faults, zoom-ins on the strain type and time evolution diagrams of the incremental
and cumulative strain types of selected points of models 2 and 3 are provided in Fig. 7. For model 2, points (a) and (b) (upper
left-hand panel in Fig. 9) are positioned in the footwall of the thrust fronts of thrust systems II and III within the eastern basin,
respectively. The evolution of the two principal stretches λ (Hencky strains) shows increasing shortening at the thrust front
after 50 min in model time for point (a) and slightly later (80 min) for point (b) (middle column in Fig. 7a-b). The dominant
strain type for both points (a) and (b) is shortening (dilatation < 1, see right column in Fig. 7a-b), but the incremental strain
type plots in between strike-slip and shortening/extension (right column in Fig. 7a-b), indicating transpressive to transtensive
motion along the thrusts. For point (a), the temporal evolution of the incremental strain type deciphers a change from thrusting
to strike-slip to extension (right column of Fig. 7a); for point (b) transpression until approximately 140 min in model time,
leading into pure shortening. Comparable to model 2, points (c) and (d) of model 3 (middle left-hand panel in Fig. 7), which



are positioned at the thrust fronts of thrust systems II and IV within the eastern basin, respectively, show shortening as dominant strain type with increased oblique slip motion of point (c) towards the end of the model run.


**[Figure 7]**

### 4.2.2 Influence of basal décollement rheology

When combining an obliquity of 20° between pre-existing structures and the shortening direction, and a variation of basal
décollement strength and behaviour, differences in (i) the number of thrust sheets forming and (ii) style of fault reactivation are observable. Model 7 (Fig. 8a-c) shows oblique (20°) basin inversion of an entirely brittle model where a 0,6 cm thick layer of glass beads below the brittle crust of quartz sand simulates the presence of a frictional décollement (Table 1, Fig. 4a). Model 8 (Fig. 8d-f) represents oblique (20°) basin inversion of a brittle-ductile model with a 0,6 cm thick viscous layer below the brittle crust (Table 1, brittle/ductile setup in Fig. 4a). Both models 7 and 8 have a platform-basin thickness ratio between 0,7
and 0,8 (i.e., underfilled basin) (Table 1). Notably, the basins are wider and more normal faults developed in case of a viscous décollement (compare Fig. 8c with 7f). As such normal faults reach farther into the platforms (e.g., western platform in Fig. 8d-e) and are mostly parallel but curved and offset across strike through relay ramps (Fig. 8d-e).

In map-view, model 7 (Fig. 8a-b) and model 8 (Fig. 8d-e) both show curved thrust fronts. At an early stage of deformation (i.e., thrust sheet I of model 7 and 8, Fig. 8a-b, d-e), the thrust front undulates but is approximately at the same distance from
the backstop, for the platform as well as basin areas. At the final stage of deformation (i.e., 9 cm of shortening, Fig. 8a-b, d-e) the thrust front steps back in basinal areas and is located further towards the foreland on platforms. Comparable to oblique inversion models 2 and 3, the strike of the thrusts where curvature is maximum is at angles of 55 to 60 degrees with respect to the shortening direction (e.g., transition of platform hosted thrust 4 to basin hosted thrust 5 of model 7 within the eastern basin in Fig. 8a-b). Deformation is concentrated at eastern borders of the respective basins, independent of variations in material at
the base of the model (Fig. 8a-b, d-e). This is shown by obliquely striking thrusts evolving on the eastern platform and connecting with basin hosted thrusts at the eastern border of the eastern basin (Fig. 8a-b, d-e).

**[Figure 8]**

Lateral ramps and transfer zones of platform thrusts to basin thrusts strike oblique, the setting being mostly compressional (red colours in cumulative strain maps in Fig. 8b, e). However, the compressional domain is accompanied by slight oblique slip movements, as zoom-ins on the strain type and time evolution diagrams of the incremental and cumulative strain types for selected points of models 7 and 8 show (Fig. 9). For model 7, points (a) and (b) (upper left-hand panel in Fig. 9) are positioned in the footwall of the thrust front of thrust system II. The actual thrust front location is shown through areas of strongest
convergence of black material lines in the zoom-in panels of the strain type (left column in Fig. 9a-b). The evolution of the





two principal stretches shows increasing shortening at the thrust front after 115 min in model time for point (a) and slightly earlier (105 min) for point (b) (middle column in Fig. 9a-b), the latter located (i) closer to the moving wall and (ii) within the eastern basin where the model crust is thinner compared to the platform. The dominant strain type for both points (a) and (b) is shortening (dilatation < 1, see right column in Fig. 9a-b), but the incremental strain type for point (b) plots in between strike-

slip and shortening (right column in Fig. 9b), referring to lateral variations in oblique slip along the thrust front of thrust system II and slightly higher oblique slip within the eastern basin (towards the eastern platform) than on the western platform. The difference in the amount of lateral oblique slip applies to the thrust front of model 8 (points (c) and (d) in the middle left-hand panel in Fig. 9), where point (d), located at the footwall of the frontal thrust of thrust system II within the eastern basin, shows most of the time purely strike-slip motion (right column in Fig. 9d), underlined by an equal deformation magnitude of the two

principal stretches (middle column in Fig. 9d). Instead, point (c), located on the western platform, shows mostly shortening, but an incremental strain type plotting between strike-slip and shortening, referring to slight oblique slip strongly overprinted by shortening (middle and right columns in Fig. 9c). The onset of deformation in brittle-ductile model 8 is, for points (c) and (d), right at the model start (middle column in Fig. 9c-d).

When using glass beads or a viscous layer as basal décollement instead of quartz sand only, two major thrust systems (Fig. 8)

instead of three (model 1, Fig. 5) or four to five thrust systems (model 2 and model 3, Fig. 6) form. Both, model 7 and model 8, show an initial thrust system I with a W-E strike covering the entire width of the models. Thrust system II of model 7 has its eastern termination within the eastern basin, whereas thrust system II of model 8 extends from the western basin towards the eastern platform. In map view, the style of deformation and thrust system evolution of model 7 is comparable to that of model 3 (Fig. 6d-e), but with longer (by ca. 1 cm) ramps and flats (compare cross-sections in Figs. 6 and 7).

In cross-sectional view, normal faults dip shallower and are slightly more listric in style of the normal faults compared to using quartz sand only (compare Fig. 5 and Fig. 6) or a viscous layer below quartz sand (Fig. 8d-f). Therefore, decreasing dip-angles towards lower parts of the faults in combination with 20° of obliquity between inherited structures and shortening direction lead to reactivation of normal faults at very low angle, most likely along W dipping normal faults and in fault segments close to the backstop where the orogen is at maximum height (thrusts 1-5, 7 in section b-b' in Fig. 8c). Due to the initial phase of

extension, the layer of glass beads (model 7, Fig. 8a-c) and silicon putty (model 8, Fig. 8d.f) thins out in basinal areas (sections b-b' and c-c' in Fig. 8c, sections e-e' and f-f' in Fig. 8f) and normal faults steepen and show a decrease in listric behaviour towards areas of lower glass beads and silicon putty thickness, respectively (sections b-b' and c-c' in Fig. 8c, sections e-e' and f-f' in Fig. 8f). This is directly related to a limited amount of reactivated normal faults as thrust faults in models 7 and 8. In model 7 only the lowest fault segment of thrusts 1 and 3-5 represents reactivation of an inherited normal fault; in model 8 only

the lowest fault segment of thrust 5. Instead, both in models 7 and 8, normal faults mostly get folded and cross-cut by thrust faults as their dip angle mostly exceeds 45° and their dip direction varies between W- and E-dipping (section b-b' in Fig. 8c and section e-e' in Fig. 8f).

**[Figure 9]**




For model 8, the area of deformation affected by shortening is comparable to brittle-only models, but the viscous layer facilitates transfer of deformation to the external part of the thrust system where inversion of normal faults takes place (e.g., in the footwall of for-thrust 5 in section e-e' in Fig. 8f). Other than in brittle-only models, where deformation jumps towards the foreland when a specific thickness of the thrust sheet is reached, in model 8 shear zones close to the backstop within the

ductile layer lead to polyphase reactivation of thrust sheet I (thrust 1 in section d-d' in Fig. 8f, thrusts 1-4 in section e-e' in Fig. 8f). Because of imbrication within the thrust sheet I and simultaneous growth of the orogen, the cumulative strain type of thrust sheet I is strongly dominated by extension (Fig. 8e).

In general, the thrusts evolved in both experiments in-sequence, with a vergence varying from mostly pop-up structures using a ductile layer (sections d-d', e-e' in Fig. 8f) to foreland-directed using glass beads as basal detachment (sections a-a', b-b' in

Fig. 8c).

### 4.2.3 Influence of basin fill rheology

In models 11 and 12, glass beads and feldspar sand represent the basin fill, respectively, instead of quartz sand (model 4). Models 4, 11, and 12 show that a platform-basin thickness ratio of 1 basically leads to smaller differences in wavelengths of thrusts on platforms and in basins compared to experiments with platform-basin thickness ratios of 0,7 to 0,8 (Table 1, model

1 in Fig. 5, models 2 and 3 in Fig. 6, models 7 and 8 in Fig. 8).

For models 11 and 12, lateral differences in thrust orientations (e.g., curved fore-thrust 4 in model 11 in Fig. 5d-e and 5 in model 12 in Fig. 5g-h) are similar to other oblique inversion models shown in this study (e.g., models 2 and 3 in Fig. 6, models 7 and 8 in Fig. 8), with the strike of the thrusts where curvature is maximum being at slightly higher angles of 70° with respect to the shortening direction (e.g., transition of platform hosted thrust 4 to eastern basin of model 11 in Fig. 10d-e). Instead, in

model 4, thrusts do not change their orientation laterally across platform boundaries. This suggests that, when using the same thickness of quartz sand both for platforms and basins, the strength difference along the normal faults is not sufficient to produce changes in thrust orientation, but you also need a weaker basin fill, as glass beads of model 11 or feldspar sand of model 12 as basin fill simulate.

Ramps of thrust faults show similar extents on platforms and in basins (e.g., 4 cm ramp length of platform hosted thrust 8 of

model 4 in section a-a' in Fig.9c compared to 4 cm ramp length of basin hosted thrust 10 of model 4 in section b-b' in Fig. 10c), whereas flat parts of thrust faults are slightly shorter in basinal realms (e.g., 3 cm flat length for platform hosted thrust 5 of model 12 in section i-i' in Fig. 10i compared to 2 cm of flat length of basin hosted thrust 8 of model 12 in section k-k' in Fig. 10i) or are partly not properly visible due to stronger imbrication in basinal realms. Exceptions are long flats in basinal realms of models 4 and 11 above sets of non-reactivated normal faults (e.g., 5 cm of flat length of thrust 8 of model 4 in section

c-c' in Fig. 10c and 4 cm of flat length of thrust 8 of model 11 in section g-g' in Fig. 10f).

**[Figure 10]**



Deformation localises at the position of pre-existing normal faults and contractional faults grow from there (e.g., thrusts 1-6

in section b-b' in Fig. 10c, thrusts 1-9 in section f-f' in Fig. 10f, thrusts 1-8 in section j-j' in Fig. 10i). Concentration of deformation appears again preferred at eastern borders of basins, i.e., where thrust faults of the younger deformation phase interact with and partly reactivate pre-existing, mostly W-dipping, normal faults (e.g., fully reactivated normal faults as thrust faults 4, 6, 8, 10 in section f-f' in Fig. 10f or thrust faults 5-8 in section j-j' in Fig. 10i). Normal faults of model 4 are mostly cross-cut (e.g., by fore-thrusts 1 to 6 in section b-b' or by fore-thrusts 7-8 in section c-c' in Fig. 10c), as normal faults dip

slightly steeper in model 4 (quartz sand basin fill; section d-d' in Fig. 10c) compared to model 11 (glass beads basin fill; section h-h' in Fig. 10f) and model 12 (feldspar sand basin fill; section l-l' in Fig. 10i). The lower friction coefficient of glass beads compared to quartz sand and feldspar sand lead to shallower (average of 55°) dipping faults (normal and reverse faults) (model 11; sections f-f', g-g', and h-h' in Fig. 10f). However, normal faults get reactivated most in model 12 (sections j-j' and k-k' in Fig. 10i), where fully reactivated normal faults as compressional faults reach the model surface within thrust system II (thrusts

6-8 in section k-k' in Fig. 10i).

Fault reactivation and more pronounced foreland transport along major thrust faults (i.e., a wider orogen in N-S direction) are characteristic mechanisms for models incorporating glass beads as basin fill (model 11; Fig. 10d-f), where the growth of the orogen in height is lower compared to models 4 and 12. Fault reactivation and orogen growth in height are the dominating mechanisms when using feldspar sand as basin fill (model 12; Fig. 10g-i). The overall deformation style is in-sequence for all

experiments with a platform-basin thickness ratio of 1, mostly foreland directed when using glass beads and feldspar sand; using quartz sand as basin fill leads to a combination of foreland directed and pop-up structures.

## 5 Discussion of modelling results

In the following sections, we summarise (Fig. 11) and discuss the experimental results and compare them with previous studies.

### 5.1 Summary of modelling results

Our experiments show that the style and orientation of contractional structures is strongly affected by the inherited rift geometry and the rheology of the basin fill. In particular, orientations of thrust fronts vary laterally across the inherited structures in all models, except for model 4 (Fig. 10a), where the strength difference of the materials used for platforms and basins (i.e., quartz sand) was too low. The oblique strike of thrusts across platform boundaries is accompanied by slight oblique slip along thrust faults and reactivated normal faults as thrust faults in oblique inversion models, shown by temporal evolution

of incremental strain at key locations in selected models (e.g., Figs. 7 and 9).

Models where shortening was oblique to the rift axes (10 and 20 degrees) and the platform-basin thickness ratios less than 1 lead to (i) a shift of the transfer zone of thrust faults connecting platform with basin realms from basin centers (e.g., parallel inversion of model 1 in Fig. 5) to platform-basin transitions, and (ii) a marked variability of thrust strikes of up to 70° with





respect to the shortening direction (compared to an average of 37° for parallel inversion models). Additionally, such kinematic boundary conditions favour the reactivation of W-dipping normal faults (e.g., normal faults 4, 6-7 in section b-b' in Fig. 6c, normal faults 4 and 7 in section g-g' in Fig. 6f) whereas E-dipping normal faults are preferentially cut by contractional structures leading to thrusting of platforms on top of basin sequences and an increase of ramp lengths. Ramps and flats are especially shorter in basins in parallel inversion models as basins consist initially of thinner crust (e.g., compare section b-b' of parallel inversion model 1 in Fig. 5g with section b-b' of oblique (10°) inversion model 2 in Fig. 6c).

Models including a frictional basal décollement result in (i) shallower dipping normal faults with an average dip of 55° compared to models without (average dip of 63°), (ii) fewer thrust systems (two instead of three or more in models without basal décollement), (iii) longer ramps and flats especially on platforms, and (iv) fewer back-thrusts. Models with a viscous basal décollement show (i) curved and through relay ramps offset normal faults (ii) deformation spreading over larger areas due to distribution of deformation within the ductile layer, (iii) normal faults not incorporated in the fold-and-thrust belt

experiencing reactivation as thrust faults (e.g., reactivated normal fault in the footwall of fore-thrust 5 in section e-e' in Fig. 8f), and (iv) fewer thrust systems similar to models with a frictional basal décollement.

Models with higher platform-basin thickness ratios of 1 and variable material for the basin fill (i.e., other than quartz sand) result in (i) more thrusts (e.g., up to 10 thrust faults in the eastern basins of model 4 and 11 in section b-b' of Fig. 10c and section f-f' in Fig. 10c, respectively), (ii) even narrower spacing of thrusts within basins compared to basins of other oblique

(20°) inversion models presented in this study (e.g., model 3 in Fig. 5d-f and models 7-8 in Fig. 8), and (iii) lateral variations of thrust orientations across platform-basin boundaries.

**[Figure 11]**

**5.2 Do inherited extensional structures trigger strain localisation during contraction?**

Basins in general, as well as normal faults at platform borders, represent natural weakness zones in which deformation concentrates during parallel to oblique basin inversion (Doglioni, 1992; Munteanu et al., 2013). Results of all models presented in this study confirm that deformation localises in areas of lateral strength contrasts in the crust such as transitions from platforms to basins, which are characterised by intense faulting and a change from basement or platform to basin sequences

(Figs. 5, 6, 8, 10, 11). With respect to the former, friction is decreased by about 17% for the quartz sand wherein the normal faults developed as shown by ring-shear experiments inferring frictional properties for peak- and reactivation conditions (Willingshofer et al., 2018). A friction coefficient of 0.52 for fault zone material falls thus within the possible range of friction values for reactivating normal faults dipping at 60° (Sibson, 1995). In contrast, analogue modelling studies using materials with higher reactivation friction reported less evidence of fault reactivation (Panien et al., 2005).

Previous studies have furthermore shown that localization of deformation through the reactivation of pre-existing faults, is favoured when the shortening direction is at angles smaller than 45° with respect to the strike of the inherited discontinuities





(e.g., Nalpas et al., 1995; Brun and Nalpas, 1996; Amiliba et al., 2005; Panien et al., 2005; Del Ventisette et al., 2006; Yagupsky et al., 2008; Deng et al., 2020) or the fault is substantially weakened by elevated pore-fluid pressure (Sibson, 1995). Yet the latter is not part of our experimental work, our modelling results support these earlier findings as demonstrated by the fact that

inherited normal faults are more often reactivated upon oblique inversion. Additionally, we note that normal faults dipping against the direction of shortening (W-dipping normal faults in our models) seem to be better oriented for reactivation than E-dipping normal faults (Figs. 6, 8, 10). Consequently, E-dipping normal faults are preferentially cut by newly formed thrust faults. A similar relationship has been described by Panien et al. (2005).

Lateral strength variations caused by the transition from rigid platforms to the weak basin fill is supported through tighter

spacing and therefore a higher number of in-sequence thrusts (e.g., compare platform section e-e' and basin sections f-f' and g-g' of model 11 in Fig. 10f). These results are consistent with earlier modelling studies demonstrating the importance of strength variations in the crust for the localisation of deformation (Brun and Nalpas, 1996; Sokoutis and Willingshofer, 2011; Bonini et al., 2012; Calignano et al., 2015; Auzemery et al., 2020).

### 5.3 Are the vertical motions at platforms different to basins?

The style of thrust faulting is overall comparable on platforms and in basins and mostly in-sequence, the latter is shown as the preferred deformation style in many previous analogue modelling studies (e.g., Ellis et al., 2004; Panien et al., 2005; Deng et al., 2020). Tighter spacing of thrusts in basinal areas, mostly depending on the difference of initial crustal thickness (Mulugeta, 1988) (i.e., platform-basin thickness ratio of 0,7-0,8 in all but models 4, 11, and 12 where it is 1), accompanied with shorter and steeper ramps, does not lead to enhanced vertical motions within basinal areas. Instead, longer ramps and flats on platforms

result in higher topography, also at the final stage of the inversion models. The rheology of the basal décollement is of importance regarding differences in vertical motion across basin boundaries. Using, e.g., a viscous basal décollement leads to shearing within the ductile layer close to the backstop within the eastern basin of model 8 (section e-e' in Fig. 8f) and therefore to strong uplift of thrust sheet I. Overall, models including a basal décollement (i.e., frictional or viscous) show lower vertical motions than models without (Liu et al., 1992; Ravaglia et al., 2006), independent of platforms or basins.

Regarding basins, we modelled different basin sizes to conceptually test variations in fault localisations. Large and small basins with ranges of basin sizes between 11,0-14,0 cm, average 12,2 cm (i.e., western basin) and 8,2-9,5 cm, average 8,6 cm (i.e., eastern basin), respectively (Fig. 12a). The style and number of thrust faults (e.g., 6 major fore-thrusts within both the western and eastern basins of model 11 in Fig. 10d-e) are very similar in large and small basins. Comparing vertical motions in large and small basins, parallel inversion models (e.g., model 1 in Fig. 5) show similar vertical motions in both basins, whereas

oblique inversion leads to enlarged vertical motions within the eastern basin. Due to oblique inversion, western platforms get thrusted onto basin successions of eastern basins (e.g., models 2 and 3 in Fig. 6, models 7 and 8 in Fig. 8, models 4, 11, and 12 in Fig. 10), resulting in successive gain of crustal thickness compared to the western basin.

**[Figure 12]**




## 5.4 What controls the variation in strike directions of the major thrust faults?

Our analogue models emphasize strong lateral variability in thrust fault orientation across platform-basin transitions (i.e., pre-existing rheological discontinuities), which has also been observed by Ravaglia et al. (2004); Di Domenica et al. (2014). The lateral variability of thrust fault strikes can be up to 70° with respect to the shortening directions. The close correlation of these
variations with the platform-basin transitions suggest causal relationships with the orientation of inherited strength variations. We also note that this feature is a robust model outcome and not applicable for the exceptional case where the basin is completely filled with material of the same strength as the platforms are made of.

Quantitative differences of thrust front positions from the backstop in both western and eastern basins are presented in Fig. 12b. From a conceptual point of view, deformation localises in both, large and small basins, but backstepping of the thrust
front is more profound in larger basins (Fig. 12b) due to more space for lateral ramps. This applies to stronger undulations of thrust faults in experiments where the thrust front is located further towards the hinterland within the large basin (i.e., western basin) (Fig. 11, Fig. 12b). In other cases, where the thrust front in the large and small basin are located at equivalent distances from the backstop (e.g., models 4, 7, 11 in Fig. 12b), the thrust front is either continuous from the western to the eastern basin or an additional fore-thrust formed separately in the western basin without connection towards the eastern basin (e.g., model
7 in Fig. 11m-n). In model 4, the location of the thrust front within the western basin is even further in the foreland of the orogen compared to the western platform (Fig. 11g-h, Fig. 12b), what could be (i) influenced from the presence of another platform to the west, as a wider backstop was used in this experiment in order to conceptually test basin inversion in large basins with straight graben borders or (ii) due to the use of the same material for platforms and basins (i.e., quartz sand) additional to a platform-basin thickness ratio of 1, resulting in a too low strength contrast between platforms and basins.
Earlier studies including strength contrasts between basins and surrounding areas by variations in crustal thickness and in basin fill material yielded similar results (Nalpas et al., 1995; Panien et al., 2005) suggesting that lateral heterogeneities within the model already prior to the shortening phase and are a major controlling factor for the undulation of the thrust faults. As such, their nature is different to variations of thrust orientations related to lateral variation of décollement strength (Cotton and Koyi, 2000; Nieuwland et al., 2000). Similar to Nalpas et al. (1995); Panien et al. (2005) we tested the influence of basin fill rheology
on the evolution of shortening structures. The results consistently indicate the mechanical stratification of the basin fill exerts a strong control on the style of deformation and the orientation of the shortening structures. On the scale of the lithosphere, Calignano et al. (2017) show that pre-existing heterogeneities that are oblique to the shortening direction can lead to the formation of oroclines.

Surprisingly, the above described transfer zones, the oblique thrust segments that connect basins and platforms, show little
evidence for strike-slip movement (Figs. 7 and 9). We suggest, that this can be explained by strain partitioning as described in complex fault systems (Krstekanić et al., 2021; Krstekanić et al., 2022).



## 6 Application to polyphase deformation within the Dolomites Indenter of the eastern Southern Alps

The model outcomes show that the presence of an inherited platform-basin configuration controls the localisation and overall style of deformation during the subsequent shortening phase. These first-order results of our crustal-scale analogue modelling

study agree with previous studies of the Dolomites Indenter of the eastern Southern Alps, highlighting the importance of inherited Mesozoic structures on Alpine deformation (Doglioni, 1992; Schönborn, 1999; Verwater et al., 2021). Other than in previous analogue modelling studies where indenters were assumed rigid (Tapponnier et al., 1982; Ratschbacher et al., 1991; Luth et al., 2013a; Luth et al., 2013b; Krstekanić et al., 2021; Krstekanić et al., 2022) we focus on indenter internal deformation and therefore follow Sokoutis et al. (2000); Willingshofer and Cloetingh (2003); Van Gelder et al. (2017), stating indenters

are never completely rigid. Kinematically, the model configuration of oblique (20°) basin inversion comes closest to SSE-directed inversion of approximately N-S striking inherited discontinuities within the Dolomites Indenter of the eastern Southern Alps.

### 6.1 Best fit model for the Dolomites Indenter

Evaluating all conducted experiments (Fig. 11), the outcome of model 8 (Table 1, Fig. 8d-f) resembles most closely the natural

example of the Dolomites Indenter, particularly on aspects of (i) style of extensional structures, (ii) overall style of the subsequent shortening phase and (iii) shortening within basinal areas. Not included in any of our crustal-scale analogue models and therefore also not in the best-fit model are (i) a mechanically stronger northern part of the Trento platform, related to the presence of the up to 2 km thick Athesian Volcanic Complex (Bosellini et al., 2007) and (ii) the presence of the sinistral transpressive Giudicarie fault system delimiting the Dolomites Indenter to the NW (Castellarin and Cantelli, 2000; Viola et

al., 2001; Pomella et al., 2012; Verwater et al., 2021), which strikes slightly oblique to parallel to Late Triassic/Jurassic extensional structures and oblique to Neogene compressional structures.

Extensional structures of the best-fit model are characterised by curved fault segments, which are connected via relay ramps. In this model normal faulting also affected parts of the Trentino platform resulting in tilted fault geometries and half-grabens (Fig. 8d-f). These structures are in accordance with local to regional scale graben structures within the platforms of the

Dolomites Indenter, e.g., the Seren graben (Doglioni, 1992; Doglioni and Carminati, 2008; Sauro et al., 2013) located within the hanging wall of the Bassano-Valdobbiadene thrust and probably controlled by inherited Jurassic geometries. During subsequent shortening those graben structures partly get reactivated and inverted, both in analogue models (section e-e' in Fig. 8f) as in the natural analogue (Sauro et al., 2013).

The in-sequence deformation style of the shortening phase fits well to the documented in-sequence thrust sequence of the

Southern Alps (Doglioni, 1992; Castellarin and Cantelli, 2000). The pop-up structure of thrust system II on the western platform of the best-fit model (section d-d' in Fig. 8f) is in line with, e.g., the so called Asiago (i.e., Sette Comuni) pop-up structure between the Bassano-Valdobbiadene fore-thrust and the Val di Sella back-thrust (Fig. 1b) (Barbieri, 1987; Barbieri and Grandesso, 2007) of the natural analogue Trento platform. The Asiago pop-up is documented as wide box-fold becoming



narrower when entering the Belluno basin and ending in a transpressive way at the transition of the Belluno basin to the Friuli
platform (Doglioni, 1990, 1992). A decrease in size of the pop-up structure from the Trento platform towards the Belluno basin
is also documented in the best-fit model (compare sections d-d' and e-e' in Fig. 8f). Comparing the style of thrusting on the
platform (section d-d' in Fig. 8f) and in the basin (section e-e' in Fig. 8f), ramps show shallower dips in basinal compared to
platform successions, resulting in longer ramps on platforms and shorter ramps in basins, taking thicker model crust on
platforms compared to basins into account. Flats in basins show different positions compared to flats on platforms (lower
height and closer to the backstop). This observation is in accordance with models of Doglioni (1992), where, e.g., the anticline
in the hanging wall of the Bassano-Valdobbiadene thrust (Fig. 1b) is located further external on the Friuli platform (i.e.,
Maniago thrust in Doglioni (1992)) than within the Belluno basin (i.e., Bassano thrust in Doglioni (1992)), across the W-
dipping normal fault transition zone at the transition from Friuli platform to Belluno basin. W-dipping normal faults show,
according to Doglioni (1992), especial strong sinistral reactivation (e.g., W-oriented faults within the sinistral transpressive
Giudicarie belt at the margin of the Trento platform towards the Lombardian basin). In contrast, our models suggest strike-slip
movement (mostly transpression) of (reactivated normal) faults at western boundaries of basins (i.e., along E-dipping normal
faults) (Fig. 9c-d). This is in accordance with strike-slip transpressive reactivation of paleostructures oriented oblique to the
shortening direction along lateral ramps (Schönborn, 1999), like, e.g., the Cimolais-Longarone or Tagliamento zones
(Nussbaum, 2000). The Cimolais-Longarone zone (Nussbaum, 2000), e.g., is located within the Belluno basin, in the hanging
wall of the Belluno thrust, directly north of the transfer zone of the Bassano-Valdobbiadane thrust from Belluno basin to Friuli
platform (Fig 1b).

In terms of normal fault reactivation, best-fit model 8 shows stronger inversion of shallow W-dipping normal faults (e.g., thrust
5 in section e-e' in Fig. 8f), whereas E-dipping normal faults have more likely been folded and/or cut by compressional
structures (e.g., folded and cut normal fault in the hanging wall of fore-thrust 4 in section d-d' and in the hanging wall of fore-
thrust 1 in section e-e' in Fig. 8f). Lateral changes in fault reactivation are common at platform-basin transitions, e.g., thrust 7
in section e-e' in Fig. f, which is independent from the normal fault in its immediate footwall; the latter getting reactivated as
thrust fault straight E of section e-e' (Fig. 8d-e).

The overall style of compressional deformation documents undulations of the frontal thrust and of the fold axes of frontal
growth folds across lateral discontinuities (e.g., platform hosted thrust 2 in Fig. 8d-e and section d-d' in Fig. 8f stepping back
towards the eastern basin), the latter representing anisotropies, like lateral ramps of transfer zones in, e.g., the upper to middle
crust. Similar effects have been shown by Ravaglia et al. (2004), where growth folds in transfer zones produce lateral
culminations in the folded structures. The style of deformation within the basinal areas of best-fit model 8 is especially well in
line with the natural analogue when comparing the model cross-section to geological cross-sections through (i) the Venetian
Alps, where platform (i.e., Trento platform) get thrusted over basinal (i.e., Belluno) successions (Doglioni, 1992; Schönborn,
1999) or (ii) the Friuli Alps, where basinal (i.e., Belluno and/or Slovenian basin) sediments are located north of platform (i.e.,
Friuli platform) successions and get thrusted over the latter (Kastelic et al., 2008; Ponton, 2010). Characteristic for the Friuli
Alps is the steep to the S dipping backthrust (Fella fault), which is cut by several N-verging thrust faults (Merlini et al., 2002;



Galadini et al., 2005; Kastelic et al., 2008; Ponton, 2010; Poli and Zanferrari, 2018). In our best-fit model 8, the position of the fore-thrusts 1-4 (section e-e' in Fig. 8f) indicates polyphase shearing within the ductile layer close to the backstop (section
e-e' in Fig. 8f), supporting the crustal-scale (Poli et al., 2021) high-angle backthrust in, e.g., the model of Venturini (1990). As this shear zone (section e-e' in Fig. 8f) does not reach the model surface, discussion about the amount of lateral (dextral referring to Merlini et al. (2002)) movement along the fault is not possible.

## 6.2 Structural observations along the western Belluno fault of the Valsugana fault system

Polyphase deformation within the Dolomites Indenter of the eastern Southern Alps is well known throughout its extent from
W to E (Doglioni, 1991; Carulli and Ponton, 1992; Doglioni, 1992; Polinski and Eisbacher, 1992; Caputo, 1996; Castellarin and Cantelli, 2000; Mellere et al., 2000; Venturini and Carulli, 2002; Kastelic et al., 2008; Caputo et al., 2010; Abbà et al., 2018; Poli et al., 2021). To compare analogue modelling results with internal deformation of the Dolomites Indenter, fault slip data from existing studies were compiled and supplemented with new fault slip data and shortening directions referred from strongly folded strata from the field (Fig. 13). Overall, the main deformational phases within the Dolomites Indenter since the
Late Triassic are: (i) D0 – Late Triassic to Jurassic W-E extension, (ii) D1 – the ?Cretaceous to Paleogene top S-directed pre-Adamello phase is mentioned for completeness, but until today only suspected within the eastern Southern Alps, while detected within the western Southern Alps west of the Giudicarie belt (Castellarin et al., 1992), (iii) D2 – Paleogene top SW-directed shortening, mostly thin-skinned (i.e., Dinaric phase), (iv) D3 – Miocene top S(SE)-directed shortening, mostly thick-skinned (i.e., Valsugana phase), (v) Miocene to Pliocene top S-directed shortening, which undulates from top SSW-directed to top
SSE-directed and (vi) D5 – Pliocene to Pleistocene top E(SE)-directed shortening, mostly transpressive, and with an increasing prominence towards the east (Fig. 13).

**[Figure 13]**

Variations in structural style along strike of thrusts within the Dolomites Indenter were studied by collecting measurements of planar (bedding, fault planes, S-C fabrics) and linear (fault striation, fold axes) structural elements along several major faults. Here we present structural observations along the western segment of the Belluno thrust as a case study (Fig. 14a-b).

The Belluno thrust belongs to the Valsugana fault system and represents a southern thrust splay of the Valsugana thrust, merging into the Valsugana thrust slightly east of Borgo Valsugana (location 1 in Fig. 14b). The overall strike of the
approximately 20 km long and approximately 30° to the N dipping Belluno thrust is WSW-ESE-trending (Vignaroli et al., 2020; Zuccari et al., 2021). The hanging wall of the Belluno thrust shows a prime example of a S-verging fault-propagation fold (Mt. Coppolo anticline) with a sub-vertical forelimb (Vignaroli et al., 2020). The Mt. Coppolo anticline exposes Upper Triassic to Lower Jurassic shallow water carbonates (Dolomia Principale, Calcari Grigi Group), which are thrusted onto Cretaceous to Paleogene strata (Maiolica Formation, Scaglia Variegata Formation, Scaglia Rossa Formation) (D'alberto et al.,
780 1995).



In the western segment of the Belluno thrust, close to where it merges into the Valsugana thrust (Fig. 14b), fault slip data from within the footwall of the Belluno thrust shows top SSW-directed shortening, accompanied by strongly folded strata with a mean fold axis of 301/14 (1 in Fig. 14a, for location of 1 see Fig. 14b), suggesting SSW-NNE directed shortening. Further towards the east (location 2 in Fig. 14b), fault slip data from within the footwall of the Belluno thrust mainly shows top S-

directed thrusting, with undulations towards the SSW and the SSE (2 in Fig. 14a, for location of 2 see Fig. 14b). Top S-directed thrusting is supported by shallow plunging mean fold axes of 265/05 and 091/07 within Jurassic to Cretaceous strata in the footwall of the Belluno thrust (2 in Fig. 14a, for location of 2 see Fig. 14b). Data from location 2 origin from west of major pre-existing Mesozoic discontinuities within the Trento platform and towards the western boundary of the Belluno basin, e.g., the Seren graben and the Cismon valley alignment (Sauro et al., 2013). Even further towards the east, where the Belluno thrust

crosses the boundary from the Trento platform towards the Belluno basin (location 3 in Fig. 14b), fault slip data from its footwall provides thrusting directions towards SSE, accompanied by a mean fold axis of 250/22 (3 in Fig. 14a, for location of 3 see Fig. 14b).

**[Figure 14]**


**6.3 Variability of deformation styles and thrust fault orientations: Implications of modelling results for the eastern Southern Alps**

Fault geometries in map-view of the conducted physical analogue experiments (Fig. 11) show strong resemblance with the fault geometries in map-view of the natural prototype (Figs. 1b, 14b). Especially striking are the (i) tighter spacing of thrusts

in basinal areas (e.g., Belluno basin), and (ii) curved thrust fronts at platform-basin transitions, e.g., of the Belluno thrust both at the transition from Trento platform to Belluno basin (Zampieri and Grandesso, 2003) as from Belluno basin to Friuli platform or Bassano-Valdobbiadene thrust at the transition from Belluno basin to Friuli platform (Picotti et al., 2022). Observation from (i) match with field descriptions from the eastern Southern Alps, where the spacing of thrusts is tighter within the Belluno basin than on the Trento platform (Doglioni, 1991; Doglioni and Carminati, 2008). According to Doglioni (1991), both folds

and thrusts show reduced wavelengths in basinal areas in the natural prototype, supporting the tighter spacing of thrusts in basins. Our analogue models support longer wavelengths of thrusts on platforms in contrast to shorter wavelengths in basins (Figs. 5, 6, 8, 10). Transitions between variations in wavelengths of thrusts appear at platform-basin boundaries, at so called transfer zones. Our models indicate that the size of the basin (Fig. 12a) rules the width of the transfer zone and the lengths of oblique lateral ramps between platform and basin domains. Backstepping of the most external thrust front in basins compared

to platforms is, in most of the analogue experiments presented in this study (Fig. 11), more distinct in the western (i.e., Lombardian) basin than in the eastern (i.e., Belluno) basin (e.g., Fig. 5c-f). The increasing amount of thrusts in basins compared to platforms is not influenced by the basin size. Our study therefore emphasises that the presence of basins representing lateral mechanical strength variations (Fig. 4b) is most important for lateral variations of the deformational styles.



According to field observations, the shortening direction along several of the studied faults, e.g. the overall SSE-directed

Belluno thrust of the Valsugana fault system (Figs. 1b, 14b), changes along strike. In the case of the Belluno thrust, the shortening direction changes from top SSW to top SSE along strike (Fig. 14a-b). Field data therefore clearly show varying shortening directions along strike of a single fault. In map-view (Figs. 1b, 14b), this variation in shortening direction can be noticed by means of an anastomosing thrust front. Focusing on this western segment of the Belluno thrust (locations 1 to 3 in Fig. 14b), especially along the sector between locations 2 and 3, the Belluno thrust follows the platform boundary of the Trento

platform towards the Belluno basin. West of location 2 (between locations 1 and 2), the Belluno thrust is located further towards the foreland, further towards the S, whereas east of location 3, the Belluno thrust is located further towards the hinterland, further towards the N.

Field data are supported by PIV-analysed top-view pictures of different time steps during the run of various analogue modelling experiments of this study (e.g., time step between 15 and 20% of bulk shortening of model 1 in Fig. 14c; time step between 10

and 15% of bulk shortening of model 7 in Fig. 14d), showing trajectories of particles (black arrows in Fig. 14c-d) which indicate variations in moving directions (between SW to SE) at one particular time step of the model run. This change in orientation of the thrust front is a feature we see in most of our parallel to oblique basin inversion models (Fig. 11), as a result of different styles of faulting in laterally varying domains of mechanical strengths, as platforms and basins represent.

Taking this information from the field and the crustal-scale analogue models into account, our study strongly supports previous

studies by, e.g., Masetti and Bianchin (1987); Doglioni (1991, 1992); Schönborn (1999) that inherited structures, e.g., pre-existing normal faults leading to platform-basin geometries, control the style of subsequent compressional deformation and are the cause of variations in shortening directions along strike of thrust faults.

## 7 Conclusions

A series of crustal-scale physical analogue models was performed to investigate the effect of inherited extensional structures

on the style of younger compressional deformation, the latter parallel to oblique to the pre-existing structures. Based on our modelling results, we infer the following:

1. Modelling results of parallel to oblique basin inversion confirm the localisation of deformation in areas of lateral strength contrasts, as transitions from platforms to basins represent.

2. Curved thrust fronts and lateral ramps coincide with the transition from platforms to basins and are therefore

controlled by rheological changes including the weakness of inherited extensional faults and the transition to the weaker basin fill. These areas, referred to as transfer zones, are the surface expression of thrust connections from platform to basin realms. Transfer zones also involve lateral changes in shortening direction along strike of particular thrust faults.

3. Reactivation of inherited normal faults is favoured for oblique shortening and predominantly occurs on fault planes

dipping towards the shortening direction (i.e., the moving backstop).



4. Although compressive strain dominates, undulating thrust fronts across platform-basin boundaries are accompanied by minor incremental oblique slip movements, ranging from transtension to predominantly transpression.

5. Spacing of in-sequence thrusts is larger on platforms and smaller in basins, which is, together with the overall style of deformation, less dependent on (i) the material used for the basal décollement, (ii) the style of graben borders, or (iii) the size of the basin, but is controlled by the presence of inherited platform-basin configuration.

Transferred to the natural analogue, the Dolomites Indenter of the eastern Southern Alps, our results strongly suggest that the whole tectonic evolution of the Dolomites Indenter with variabilities of shortening directions along strike of several thrust faults (e.g., the Belluno thrust of the Valsugana fault system) is controlled by inherited structures and does not necessarily reflect different deformation phases. As such the number of deformation phases in the Southern Alps may have been 855 overestimated so far.

**Author contributions.** AKS – Conceptualisation, Methodology, Validation, Formal analysis, Investigation, Data curation, Writing – original draft, Writing – review & editing, Visualization. EW – Conceptualisation, Methodology, Validation, Resources, Writing – original draft, Writing – review & editing, Supervision. TK – Conceptualisation, Writing – review & 860 editing. HO – Conceptualisation, Writing – review & editing, Supervision. HP – Funding acquisition, Conceptualisation, Writing – review & editing, Supervision.

**Acknowledgements.** This research is funded by the Austrian Science Fund (FWF), project P 33272 (PI Hannah Pomella), and is part of a collaboration between the Department of Earth Sciences of the Utrecht University, Netherlands and the Department of Geology of the University of Innsbruck, Austria during the PhD project of the first author. The physical analogue 870 experiments were performed at the Tectonic Modelling Laboratory (TecLab) of the Utrecht University, where the first author benefitted from the assistance of Sjaak van Meulebrouck and of helpful discussions with Eva Bakx.

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




**Figure 1. (a) Topographic map of the Alpine Arc of western Austria, Switzerland, northern Italy and Slovenia overlain by first-order structures of the Alpine orogen (modified from Schmid et al. (2004); (2020)). Abbreviations: PG – Pusteral-Gailtal fault, GFS – Giudicarie fault system, GB – Giudicarie belt, TF – Tonale fault, CF – Calisio fault, DI – Dolomites Indenter, II – Insubric indenter. (b) Late Triassic/Jurassic platform-basin configuration (modified from Winterer and Bosellini (1981); Busetti et al. (2010); Masetti et al. (2012); Martinelli et al. (2017); Picotti and Cobianchi (2017); Picotti et al. (2022)) projected over the present day geography and overlain by the tectonic map of the Dolomites Indenter (modified from Schönborn (1999); Castellarin and Cantelli (2000); Schmid et al. (2004)). Abbreviation red: VF – Villnöss/Funes fault, WF – Würzjoch/Passo delle Erbe fault, TTF – Tremosine-Tignale fault, VB – Val Bordaglia fault, CF – Calisio fault, VdS – Val di Sella back-thrust. Abbreviation black: CadR – Cadore region, CarR – Carnia region, FriR – Friuli region, VB – Val Badia/Gadertal.**



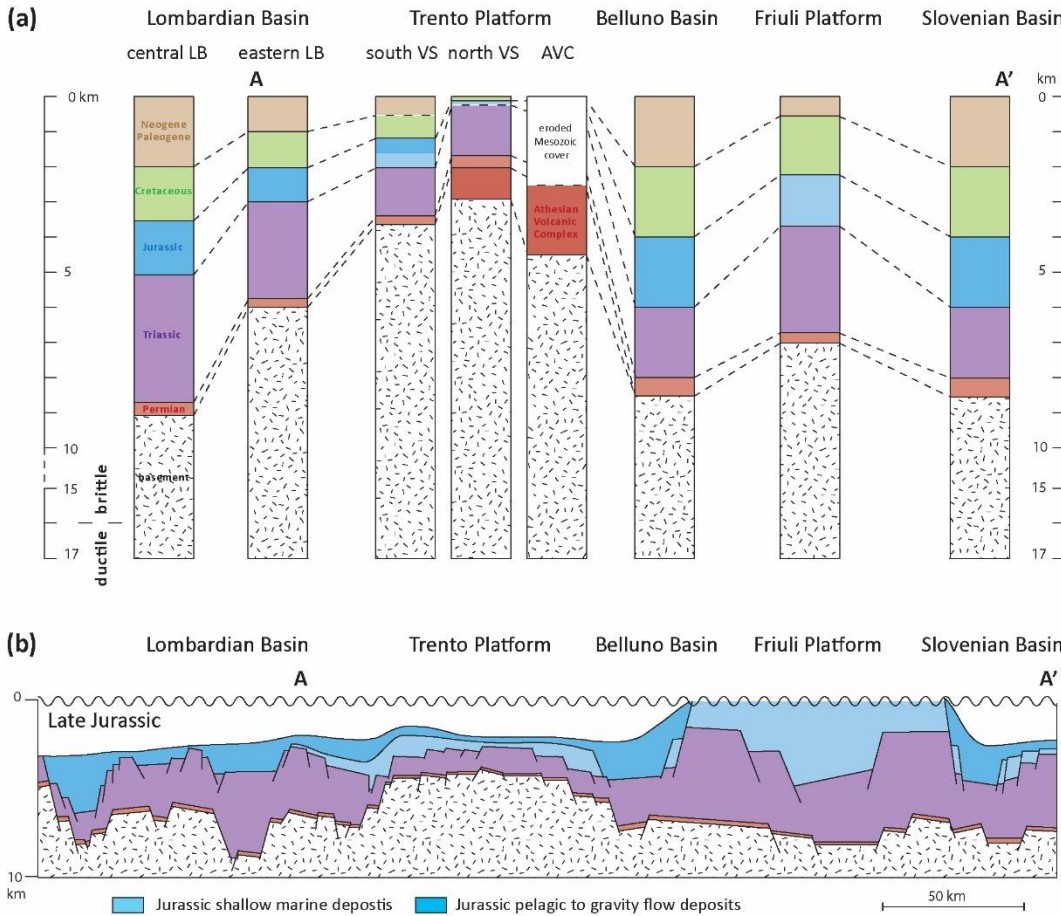

**Figure 2. Paleogeographic domains of the Dolomites Indenter. The location of cross-section A-A' is indicated in Fig. 1b. (a) Simplified stratigraphic columns for each paleogeographic domain (modified from Bertotti et al. (1993); Picotti et al. (1995); Picotti and Cobianchi (2017); Verwater et al. (2021)). (b) Cross-section (vertically exaggerated!) through the Jurassic platform-basin configuration (modified from Winterer and Bosellini (1981); Smuc and Goričan (2005)).**



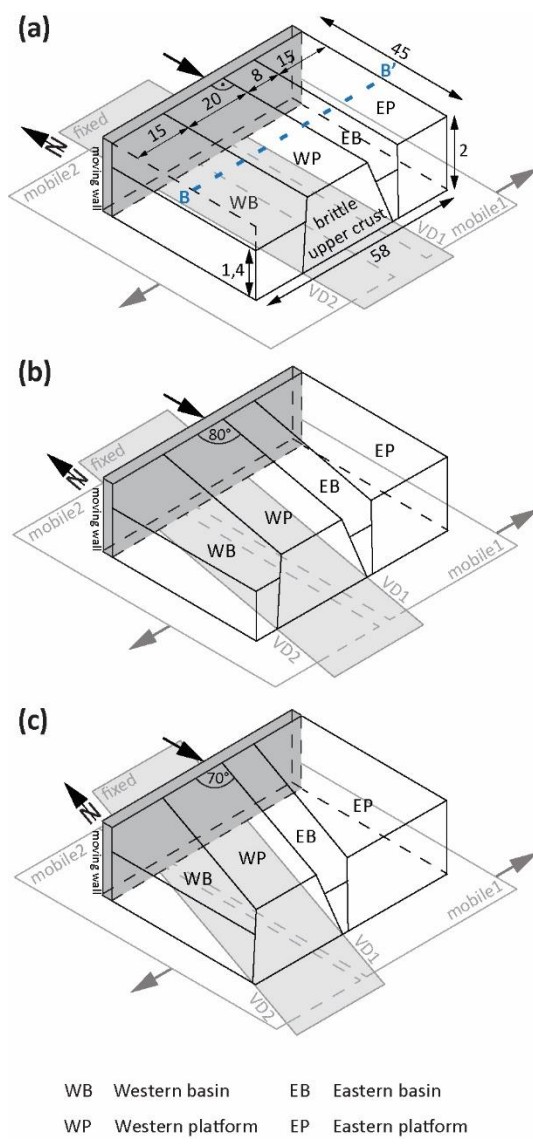

| WB | Western basin | EB | Eastern basin |
| WP | Western platform | EP | Eastern platform |


**Figure 3. Simplified sketch of the analogue modelling setup post-extension and pre-shortening for (a) parallel inversion, (b) oblique inversion with an angle of 80° between backstop and pre-existing discontinuities, (c) oblique inversion with an angle of 70° between backstop and pre-existing discontinuities. All numbers in Fig. 3a-c without units are in centimeters.**



**Figure 4. Modelling setup. (a) Simplified sketches of setup cross-sections for brittle only and brittle/ductile experiments. Cross-section location is indicated in Fig. 3a (b) Strength profiles for brittle only and for brittle/ductile experiments for platforms/filled basins (platform-basin thickness ratio of 1) and for underfilled basins (platform-basin thickness ratio of 0,7 to 0,8). All numbers in Fig. 4a without units are in centimeters.**









**Figure 5. Model 1 – reference model, parallel basin inversion, quartz sand only. (a) Interpreted top-view picture after the first phase of extension and before shortening (0% of bulk shortening). (b) Interpreted top-view picture after 2,25 cm of shortening (5% of bulk shortening). (c) Interpreted top-view picture after 4,5 of shortening (10% of bulk shortening). (d) Map of cumulative strain type after 4,5 cm of shortening (10% of bulk shortening). (e) Interpreted top-view picture after 9,0 cm of shortening (20% of bulk shortening), (f) Map of cumulative strain type after 9,0 cm of shortening (20% of bulk shortening). Visually interpreted structures of Fig. 5c, e overlay strain type plots in Fig. 5d, f. Strain colour legend corresponds to Fig. 5d, f. The transparency of areas with a strain magnitude below the 90 percentile is increased for supressing areas without significant deformation. (g) Cross-sections of the reference model at the end of the experiment. Grey dashed line marks the model topography at the end of the experiment. Black and grey layers above the topography line marks the post-kinematic sand cover. Cross-section locations are shown in Fig. 5e.**









**Figure 6. Model 2 and model 3 – oblique (10°, 20°, respectively) basin inversion; quartz sand only. (a) Interpreted top-view picture of model 2 after 9,0 cm of shortening (20% of bulk shortening). (b) Map of cumulative strain type of model 2 after 9,0 cm of shortening (20% of bulk shortening). (c) Cross-sections of model 2 at the end of the experiment. Grey dashed line marks the model topography at the end of the experiment. Black and grey layers above the topography line marks the post-kinematic sand cover. Cross-section locations are shown in Fig. 6a. Note that sections a-a', b-b' and c-c' are oriented oblique (10°) to the main structures. (d) Interpreted top-view picture of model 3 after 9,0 cm of shortening (20% of bulk shortening). (e) Map of cumulative strain type of model 3 after 9,0 cm of shortening (20% of bulk shortening). Visually interpreted structures of Fig. 6a, d overlay strain type plots in Fig. 6b, e. Strain colour legend corresponds to Fig. Fig. 6b, e. The transparency of areas with a strain magnitude below the 90 percentile is increased for supressing areas without significant deformation. (f) Cross-sections of model 3 at the end of the experiment. Grey dashed line marks the model topography at the end of the experiment. Black and grey layers above the topography line marks the post-kinematic sand cover. Cross-section locations are shown in Fig. 6d. Note that sections e-e', f-f' and g-g' are oriented oblique (20°) to the main structures.**



**Figure 7.** Temporal evolution of principal stretches and strain type during the compressional phase of model 2 (a) at the thrust front of thrust system II in the eastern basin and (b) at the thrust front of thrust system III within the eastern basin and of model 3 (c) at the thrust front of thrust system II in the eastern basin and (d) at the thrust front of thrust system IV within the eastern basin. Upper left-hand panel: strain type (final) and overview of the selected areas (a) and (b) of model 2. Middle left-hand panel: strain type (final) and overview of the selected areas (c) and (d) of model 3. Left column: zoom on the strain type including the selected grid cell (outlined in red) and neighbouring grid cells (outlined in black). Middle column: temporal evolution of the logarithm of the two principal stretches (Henky strain, blue and red line). Right column: temporal evolution of dilatation, cumulative strain type, and incremental strain type.









**Figure 8. Model 7 and model 8 – oblique (20°) basin inversion; variation in material for basal décollement (glass beads and silicon putty, respectively). (a) Interpreted top-view picture of model 7 after 9,0 cm of shortening (10% of bulk shortening). (b) Map of cumulative strain type of model 7 after 9,0 cm of shortening (10% of bulk shortening). (c) Cross-sections of model 7 at the end of the experiment. Grey dashed line marks the model topography at the end of the experiment. Black and grey layers above the topography line mark the post-kinematic sand cover. Cross-section locations are shown in Fig. 8a. Note that the cross-sections a-a' and b-b' are oriented oblique (20°) to the main shortening structures. (d) Interpreted top-view picture of model 8 after 9,0 cm of shortening (10% of bulk shortening). (e) Map of cumulative strain type of model 8 after 9,0 cm of shortening (10% of bulk shortening). Visually interpreted structures of Fig. 8a, d overlay strain type plots in Fig. 8b, e. Strain colour legend corresponds to Fig. 8b, e. The transparency of areas with a strain magnitude below the 90 percentile is increased for supressing areas without significant deformation. (f) Cross-sections of model 8 at the end of the experiment. Grey dashed line marks the model topography at the end of the experiment. Black and grey layers above the topography line mark the post-kinematic sand cover. Cross-section locations are shown in Fig. 8d. Note that the cross-sections d-d' and e-e' are oriented oblique (20°) to the main shortening structures.**





**Figure 9. Temporal evolution of principal stretches and strain type during the compressional phase of model 7 (a) at the thrust front of thrust system II on the western platform and (b) at the thrust front of thrust system II within the eastern basin and of model 8 (c) at the thrust front of thrust system II on the western platform and (d) at the thrust front of thrust system II within the eastern basin. Upper left-hand panel: strain type (final) and overview of the selected areas (a) and (b) of model 7. Middle left-hand panel: strain type (final) and overview of the selected areas (c) and (d) of model 8. Left column: zoom on the strain type including the selected grid cell (outlined in red) and neighbouring grid cells (outlined in black). Middle column: temporal evolution of the logarithm of the two principal stretches (Henky strain, blue and red line). Right column: temporal evolution of dilatation, cumulative strain type, and incremental strain type.**









**Figure 10. Model 4, model 11 and model 12 – oblique (20°) basin inversion; variation in material representing the basins fill (quartz sand, glass beads, and feldspar sand, respectively). (a) Interpreted top-view picture of model 4 after 9,0 cm of shortening (20% of bulk shortening). (b) Map of cumulative strain type of model 4 after 9,0 cm of shortening (20% of bulk shortening). (c) Cross-sections of model 4 at the end of the experiment. Grey dashed line marks the model topography at the end of the experiment. Black and grey layers above the topography line mark the post-kinematic sand cover. Cross-section locations are shown in Fig. 10a. Note that the**

**cross-sections a-a', b-b', c-c' are oriented oblique (20°) to the main shortening structures. (d) Interpreted top-view picture of model 11 after 9,0 cm of shortening (20% of bulk shortening). (e) Map of cumulative strain type of model 11 after 9,0 cm of shortening (20% of bulk shortening). (f) Cross-sections of model 11 at the end of the experiment. Grey dashed line marks the model topography at the end of the experiment. Black and grey layers above the topography line mark the post-kinematic sand cover. Cross-section locations are shown in Fig. 10d. Note that the cross-sections e-e' and g-g' are oriented oblique (20°) to the main shortening structures.**

**(g) Interpreted top-view picture of model 12 after 9,0 cm of shortening (20% of bulk shortening). (h) Map of cumulative strain type of model 12 after 9,0 cm of shortening (20% of bulk shortening). Visually interpreted structures of Fig. 10a, d, g overlay strain type plots in Fig. 10b, e, h. Strain colour legend corresponds to Fig. 10b, e, h. The transparency of areas with a strain magnitude below the 90 percentile is increased for supressing areas without significant deformation. (i) Cross-sections of model 12 at the end of the experiment. Grey dashed line marks the model topography at the end of the experiment. Black and grey layers above the topography**

**line mark the post-kinematic sand cover. Cross-section locations are shown in Fig. 10g. Note that the cross-sections i-i', j-j' and k-k' are oriented oblique (20°) to the main shortening structures.**





**Figure 11. Interpreted top-view pictures and cumulative strain type maps of the final stages of all 12 experiments after 9,0 cm of shortening (20% of bulk shortening). Visually interpreted structures of Fig. 12a, c, e, g, I, k, m, o, q, s, u, w overlay strain type plots in Fig. 12b, d, f, h, j, l, n, p, r, t, v, x. Strain colour legend corresponds to Fig. 12b, d, f, h, j, l, n, p, r, t, v, x. The transparency of areas with a strain magnitude below the 90 percentile is increased for supressing areas without significant deformation.**



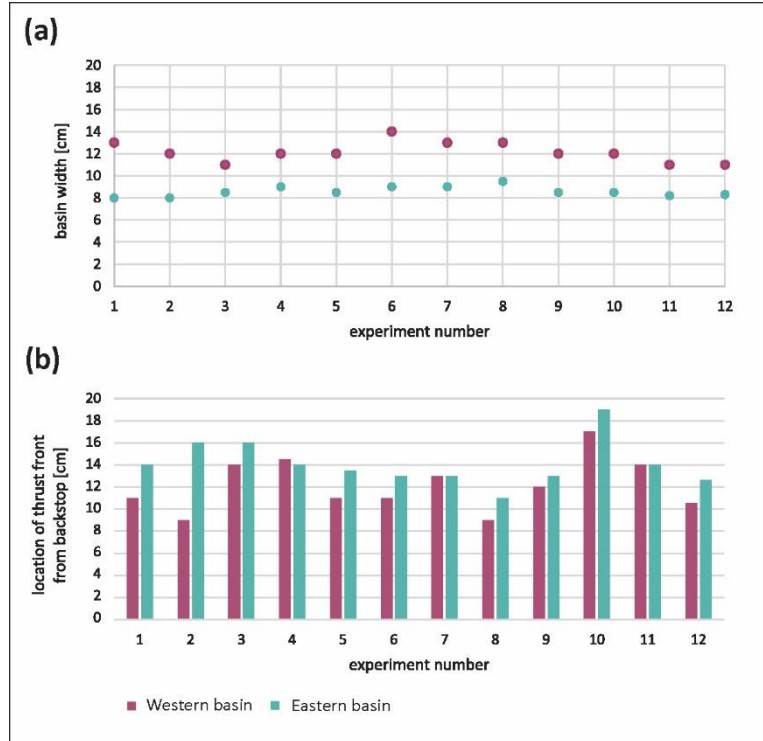

**Figure 12. (a) Chart of basin widths of the western and eastern basins of each experiment, measured at the final stage (20% of bulk shortening) of the experiment in W-E direction, perpendicular to platform boundaries. (b) Histogram showing the distance of the thrust fronts in basinal areas from the backstop, measured at the final stage (20% of bulk shortening) of the experiment on top-view photographs.**



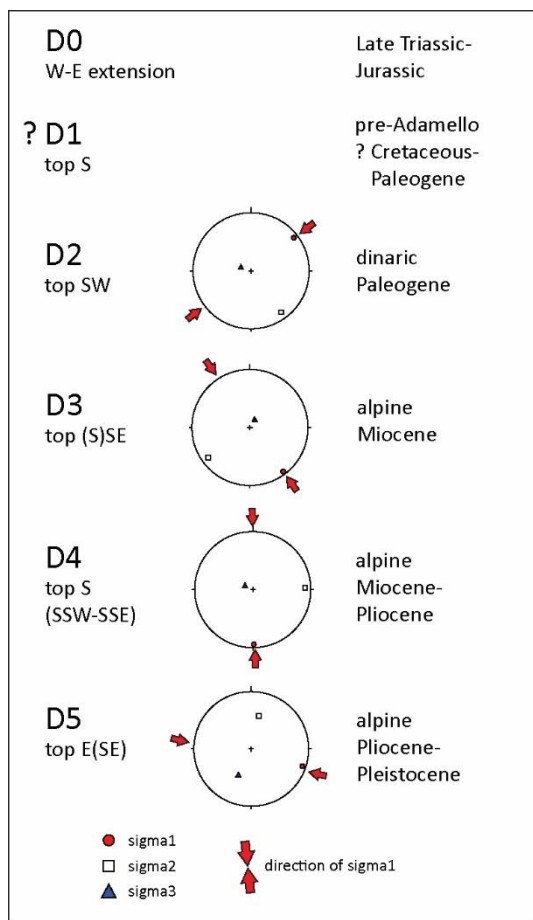

**Figure 13. Schematic overview of the main deformational phases D0 to D5 of the eastern Southern Alps (compiled from Caputo (1996); Castellarin and Cantelli (2000); Nussbaum (2000) and this study).**








**Figure 14. (a) Fault slip data with local paleostress directions and fold axes with indicated shortening directions of the Belluno thrust (Valsugana fault system). Locations 1 to 3 are indicated in Fig. 12b. (b) Late Triassic/Jurassic platform-basin configuration (modified from Winterer and Bosellini (1981); Busetti et al. (2010); Masetti et al. (2012); Martinelli et al. (2017); Picotti and Cobianchi (2017);**

**Picotti et al. (2022)) projected over the present day geography and overlain by the tectonic map of the Dolomites Indenter (modified from Schönborn (1999); Castellarin and Cantelli (2000); Schmid et al. (2004)). Note the change in shortening directions along strike of the Belluno thrust. Locations 1 to 3 represent segments of different strike of the Belluno thrust. Abbreviation red: VF – Villnöss/Funes fault, WF – Würzjoch/Passo delle Erbe fault, TTF – Tremosine-Tignale fault, VB – Val Bordaglia fault, CF – Calisio fault, VdS – Val di Sella back-thrust, CL – Caneva line. Abbreviation black: CadR – Cadore region, CarR – Carnia region, FriR –**

**Friuli region, VB – Val Badia/Gadertal, Ap – Asiago pop-up structure, Sg – Seren graben. (c) PIV analysed top-view picture of parallel basin inversion of model 1 at 18% of bulk shortening (BS). Black arrows indicate vectors of particle flow direction. (d) PIV analysed top-view picture of oblique (20°) basin inversion of model 7 at 12% of bulk shortening (BS). Black arrows indicate vectors of particle flow direction.**



**Table 1. Geometrical model parameters used in this study. Model type: B – brittle, BD – brittle-ductile.**

| Model number | Model type | Thickness brittle layer platform [cm] | Thickness brittle layer basin [cm] | Thickness ductile layer [cm] | Velocity engine pull extension [cm/h] | Velocity engine push compression [cm/h] | Total shortening [cm] | Angle of obliquity for inversion [°] | Scaled thickness platform [km] | Scaled thickness basin [km] |
|---|---|---|---|---|---|---|---|---|---|---|
| 1 | B | 2,0 | 1,4 | - | 5,0 | 3,0 | 9,0 | 0 | 16,0 | 11,2 |
| 2 | B | 2,0 | 1,4 | - | 5,0 | 3,0 | 9,0 | 10 | 16,0 | 11,2 |
| 3 | B | 2,0 | 1,4 | - | 5,0 | 3,0 | 9,0 | 20 | 16,0 | 11,2 |
| 4 | B | 2,0 | 2,0 | - | 5,0 | 3,0 | 9,0 | 20 | 16,0 | 16,0 |
| 5 | B | 2,6 | 2,0 | - | 5,0 | 3,0 | 9,0 | 0 | 20,8 | 16,0 |
| 6 | BD | 2,0 | 1,4 | 0,6 | 2,5 | 2,5 | 9,0 | 0 | 20,8 | 16,0 |
| 7 | B | 2,6 | 2,0 | - | 5,0 | 3,0 | 9,0 | 20 | 20,8 | 16,0 |
| 8 | BD | 2,0 | 1,4 | 0,6 | 2,5 | 2,5 | 9,0 | 20 | 20,8 | 16,0 |
| 9 | B | 2,0 | 2,0 | - | 5,0 | 3,0 | 9,0 | 0 | 16,0 | 16,0 |
| 10 | B | 2,6 | 2,6 | - | 5,0 | 3,0 | 9,0 | 0 | 20,8 | 20,8 |
| 11 | B | 2,0 | 2,0 | - | 5,0 | 3,0 | 9,0 | 20 | 16,0 | 16,0 |
| 12 | B | 2,0 | 2,0 | - | 5,0 | 3,0 | 9,0 | 20 | 16,0 | 16,0 |




**Table 2. Properties for brittle material used in this study.**

| Brittle material | Grain size [μm] | Density [kg/m³] | Cohesion [Pa] | Coefficient of peak friction | Coefficient of dynamic friction | Coefficient of reactivation friction |
|---|---|---|---|---|---|---|
| Quartz sand | 100-300 | 1500 | 10-40 | 0,63 | 0,48 | 0,52 |
| Feldspar sand | 100-250 | 1300 | 15-35 | 0,68 | 0,55 | 0,61 |
| Glass beads | 100-200 | 1530 | 25 | 0,48 | 0,40 | 0,44 |