# Peer review of "Inversion of extensional basins parallel and oblique to their boundaries: Inferences from analogue models and field observations from the Dolomites Indenter, eastern Southern Alps"

_EGUsphere, 2022_

## Referee Comment (RC1)

Comments on:

**Inversion of extensional basins parallel and oblique to their boundaries: Inferences from analogue models and field observations from the Dolomites Indenter, eastern Southern Alps**

By Anna-Katharina Sieberer[1], Ernst Willingshofer[2], Thomas Klotz[1], Hugo Ortner[1], Hannah Pomella[1]

General comments

The manuscript presents a series of analogue experiments meant to explore the role of pre-existing normal faults in the subsequent continental collision and thrust evolution. The experimental setting takes inspiration from the natural case of the Southern Alps. The study is well designed and the experimental part straightforward and well-illustrated. The results are of interest not only for the Alpine community, but for all Earth scientists working on the structural style of continental polyphase deformation. In my opinion, the manuscript misses a discussion between paleostress and strain partitioning on single structures. I try to explain it better in the following paragraph.

The discussion on how the reconstructed paleostress evolution correlates with the large-scale kinematic evolution, i.e. if the recorded deformation phases are due to partitioning along local structures or to regional stress field variations is long-lasting and it was addressed by several authors (e.g. Varga, 1993; De Vicente et al., 2009; Simon, 2019; Hippolyte and Mann, 2021). In the Southern Alps, the different "events" or "phases" have been recognized through regional studies with hundreds of studied sites along the whole Southern Alps and dated with a tectonic stratigraphic approach. The main picture of the paleostress evolution from the Late Oligocene to the Pleistocene was published by Castellarin et al., 1992, Castellarin et al., 1998 and Castellarin and Cantelli 2000. Later on, Caputo et al. (2010), confirmed the same paleostress evolution for the late Miocene to Quaternary at the front of the eastern Southern Alps. This research was addressed far from the main fault systems and did not consider the highly deformed rock volumes, as established by the first authors that adopted this method (e.g. Angelier, 1979; Bergerat, 1987).

Furthermore, a detailed reconstruction of the Africa-Europe convergence path, based on the magnetic anomalies of the Atlantic Ocean of Dewey (1989), presented by Mazzoli and Helman 1993 suggests a similar timing and direction of convergence as observed in the paleostress reconstruction of Southern Alps and other Mediterranean areas (see also Fig. 20 of Fellin et al., 2005 and Fig. 12 of Caputo et al., 2010).

For the exposed reasons, it is clear the authors should keep strain and stress separated in their discussions. From the authors experiments, one can visualize the role of previous faults or lateral facies juxtaposition on the inversion strain pattern. However, if the stress is local or regional, this cannot be explored by analysing few structural sites, but from much larger statistical approach. This was the Castellarin et al 1992 and Caputo et al 2010 approach. I urge the authors to develop this point in the discussion. This new discussion would bring also to a change in the last sentences of the conclusion. In fact, I think the results of this

manuscript do not question the paleostress analysis of previous authors, but it emphasizes the role of pre-existing structures in partitioning the strain along regional fault systems, such as the Belluno thrust.

In chapt. 6.2, the presentation of the various deformation phases is somehow confusing. Several issues: 1) the use of grey literature, such as the Nussbaum unpublished PhD thesis; 2) the occurrence of the pre-Adamello phase in the eastern Alps: why adding this as D2 phase, if the authors (correctly) claim this phase was never recorded in the eastern Southern Alps? 3) The authors propose Miocene (D3), a Miocene to Pliocene (D4) top S, and a final top to the E Plio-Pleistocene phase (D5). However, Castellarin and Cantelli (2000) and Caputo et al. (2010) document variable shortening axis between N340°and N310°, from Serravallian to the Pleistocene. Therefore, how did the authors separate the 3 last phases? Please, explain the method used, or correctly refer the previous authors findings
Finally, in Fig. 14 a2, the top to the S could be due to the mixing of SW and SSE directed faults, maybe formed at different time. This is a common problem of structural sites in the polydeformed Southern Alps, when cross-cutting relationships cannot be found. The authors should consider this alternative interpretation.

Specific points

Line 53 add Bernoulli and Jenkyns, 1974 in the references as the first modern paper dealing with platforms and basins in the Southern Alps.
Line 94 Early Permian (geochronology) instead of Lower Permian (chronostratigraphy).
Line 94 to 97 Actually, the Early Permian event is highly debated but very likely not associated to the rifting of the Hallstatt-Meliata ocean. See f.i. Muttoni et al. 2003 for interpretation associated to large scale intracontinental transform. Most scholars place the start of the first rifting phase in the Late Permian (e.g. Bertotti et al., 1993), ending in the Carnian.
Line 100, 134 and 722: too many brackets after Vrabeč, Zampieri, Beccaluva and Doglioni.
103-105 the quoted papers of Martinelli, Pieri & Groppi and Masetti are not suitable for the depth of faulting, since they are not based on observations, purely speculative. Two case studies of depth prolongation of the normal faults in the western Southern Alps are the Lugano fault (e.g. Bertotti, 1990) and the Pogallo fault (Handy, 1987). I would find more convincing quoting these last authors, although they are dealing with structures that slipped more than the ones of the eastern Southern Alps.
Line 136 No reference to age in Fantoni and Franciosi (2010) and Vignaroli et al. (2020)! I double checked them, since Late Oligocene SSE directed shortening is in contrast with the tectonic stratigraphy presented by Castellarin et al. (1992), Castellarin and Cantelli (2000). These latter papers, based on data collected on large parts of the Southern Alps, document a Chattian to Burdigalian SSW directed shortening on a regional scale. See General Comments.
Line 150-151 better quote Venzo 1977 for the folded Pliocene.
172 Variscan
175 Late Permian
178 Please revise this sentence. Besides the two "are", the message is not clear.
185 "bioclastic to marly sediments" is not correct for describing the Paleogene to Miocene stratigraphy. Please, use "limestone and marls".

186 During the Early Jurassic. "Footwall of rifted margins" is not correct: footwall refers to a fault limb; rifted margins refer to a larger scale epicontinental sea. You can use: "footwall of the major normal faults", instead.

193 evaporite-bearing shales, instead of facies associations.

340 which simulates

397 preferentially instead of preferred? Change e.g. (*exempli gratia*), used when you choose some examples from a larger list, with i.e. (*id est*), used when, as in this case, the authors deepen the meaning of the previous words.

445 Fig. 9 appears prior to Fig. 8, which is named first in Line 461. Consider revising the order and number of the Figures…

453 and 471 Compared to, instead of Comparable

500 Both model 7 and model 8 show. No commas

610 not clear to what are the authors referring with "former"

611 …developed, as…

676 to 678 Unclear this change of scale and the relevance of the Calignano et al 2017 paper. Please, motivate it.

688 …2022), we focus…

747 "steep to the S dipping backthrust" Please, change in: backthrust steeply dipping to the S.

771 …along major fault zones.

779 D'Alberto

936 Carnico-Friulano, 1992. Studi Geologici Camerti. Nuova Serie (1992): 275-284.

1091. Memorie di Scienze Geologiche, Padova, …

References

Angelier, J. (1979). Determination of the mean principal directions of stresses for a given fault population. *Tectonophysics*, *56*(3-4), T17-T26.

Bergerat, F. (1987). Stress fields in the European platform at the time of Africa-Eurasia collision. *Tectonics*, *6*(2), 99-132.

Bernoulli, D., & Jenkyns, H. C. (1974). Alpine Mediterranean and central Atlantic Mesozoic facies in relation to the early evolution of the Tethys.

Bertotti, G. (1990). *Early Mesozoic extension and Alpine tectonics in the western Southern Alps: The geology of the area between Lugano and Menaggio (Lombardy, Northern Italy)*(Doctoral dissertation, ETH Zurich).

Castellarin, A., Cantelli, L., Fesce, A. M., Mercier, J. L., Picotti, V., Pini, G. A., Prosser, G. & Selli, L. (1992). Alpine compressional tectonics in the Southern Alps. Relationships with the N-Apennines. In *Annales tectonicae* (Vol. 6, No. 1, pp. 62-94).

Castellarin, A., & Cantelli, L. (2000). Neo-Alpine evolution of the southern Eastern Alps. *Journal of Geodynamics*, *30*(1-2), 251-274.

De Vicente, G., Vegas, R., Muñoz-Martín, A., Van Wees, J. D., Casas-Sáinz, A., Sopeña, A., ... & Fernández-Lozano, J. (2009). Oblique strain partitioning and transpression on an inverted rift: The Castilian Branch of the Iberian Chain. *Tectonophysics*, *470*(3-4), 224-242.

Handy, M. R. (1987). The structure, age and kinematics of the Pogallo Fault Zone; Southern Alps, northwestern Italy. *Eclogae Geologicae Helvetiae*, *80*(3), 593-632.

Hippolyte, J.-C., and P. Mann, 2021, Neogene paleostress and structural evolution of Trinidad: Rotation, strain partitioning, and strike-slip reactivation of an obliquely colliding thrust belt, in C. Bartolini, ed., South America–Caribbean–Central Atlantic plate boundary: Tectonic evolution, basin architecture, and petroleum systems: Tectonic evolution, basin architecture, and petroleum systems: AAPG Memoir 123, p. 317–346. DOI: 10.1306/13692249M1233851

Muttoni, G., Kent, D. V., Garzanti, E., Brack, P., Abrahamsen, N., & Gaetani, M. (2003). Early Permian Pangea 'B'to Late Permian Pangea 'A'. *Earth and Planetary Science Letters*, *215*(3-4), 379-394.

Simón, J. L. (2019). Forty years of paleostress analysis: has it attained maturity?. *Journal of Structural Geology*, *125*, 124-133.

Varga, R. J. (1993). Rocky Mountain foreland uplifts: Products of a rotating stress field or strain partitioning? *Geology*, *21*(12), 1115-1118.

Venzo, S., with collaboration of Petrucci, F., & Carraro, F. (1977). I depositi quaternari e del Neogene Superiore nella bassa valle del Piave da Quero al Montello e del Paleopiave nella valle del Soligo (Treviso). *Memorie degli Istituti di Geologia e Mineralogia dell'Universita di Padova*, *30*, 1–27.

---

## Author Response (AR1)

**1. Point-by-point response to reviewer 1, Vincenzo Picotti**

Dear Vincenzo Picotti,

We are grateful for your valuable comments and suggestions which particularly helped improve on clarity and readability of the manuscript. All specified points have been considered, text and figures have been modified based on the comments, which we refer to in our co-listing reply below.

Please do not hesitate to reach out to us for further clarification in any of our responses to your review.

Best,
Anna-Katharina Sieberer and co-authors.

*General comments*

*The manuscript presents a series of analogue experiments meant to explore the role of pre-existing normal faults in the subsequent continental collision and thrust evolution. The experimental setting takes inspiration from the natural case of the Southern Alps. The study is well designed and the experimental part straightforward and well-illustrated. The results are of interest not only for the Alpine community, but for all Earth scientists working on the structural style of continental polyphase deformation. In my opinion, the manuscript misses a discussion between paleostress and strain partitioning on single structures. I try to explain it better in the following paragraph.*

*The discussion on how the reconstructed paleostress evolution correlates with the largescale kinematic evolution, i.e. if the recorded deformation phases are due to partitioning along local structures or to regional stress field variations is long-lasting and it was addressed by several authors (e.g. Varga, 1993; De Vicente et al., 2009; Simon, 2019; Hippolyte and Mann, 2021). In the Southern Alps, the different "events" or "phases" have been recognized through regional studies with hundreds of studied sites along the whole Southern Alps and dated with a tectonic stratigraphic approach. The main picture of the paleostress evolution from the Late Oligocene to the Pleistocene was published by Castellarin et al., 1992, Castellarin et al., 1998 and Castellarin and Cantelli 2000. Later on, Caputo et al. (2010), confirmed the same paleostress evolution for the late Miocene to Quaternary at the front of the eastern Southern Alps. This research was addressed far from the main fault systems and did not consider the highly deformed rock volumes, as established by the first authors that adopted this method (e.g. Angelier, 1979; Bergerat, 1987). Furthermore, a detailed reconstruction of the Africa-Europe convergence path, based on the magnetic anomalies of the Atlantic Ocean of Dewey (1989), presented by Mazzoli and Helman 1993 suggests a similar timing and direction of convergence as observed in the paleostress reconstruction of Southern Alps and other Mediterranean areas (see also Fig. 20 of Fellin et al., 2005 and Fig. 12 of Caputo et al., 2010).*

- **Author's reply:** Thank you for your valuable comment on including the differentiation between paleostress (regional) and strain partitioning (local) in our discussion. We very much agree with your suggestion and modified the text of the first two paragraphs of section 6.2 according to it.

*For the exposed reasons, it is clear the authors should keep strain and stress separated in their discussions. From the authors experiments, one can visualize the role of previous faults or lateral facies juxtaposition on the inversion strain pattern. However, if the stress is local or regional, this cannot be explored by analysing few structural sites, but from much larger statistical approach. This was the Castellarin et al 1992 and Caputo et al 2010 approach. I urge the authors to develop this point in the discussion. This new discussion would bring also to a change in the last sentences of the conclusion. In fact, I think the results of this manuscript do not question the paleostress analysis of previous authors, but it emphasizes the role of pre-existing structures in partitioning the strain along regional fault systems, such as the Belluno thrust.*

- **Author's reply:** Thank you for your comment on separating stress and strain in our discussion. To clarify, in Fig. 13, our data, which supplement data from previous paleostress studies, are not only the 3 field locations shown in Fig. 14a-b, but plenty more locations over the eastern Southern Alps from the Adige Valley in the west to the Friuli Alps in the east. We added this in the text of the third paragraph of section 6.2. However, we agree, that our data, although consisting of more than a few structural sites, are not comparable to the large statistical approach of Castellarin and Cantelli (2000) and of Caputo (2010). We therefore modified the text of the last two paragraphs of section 6.3 and the last paragraph of the concluding section 7 according to your suggestions.

*In chapt. 6.2, the presentation of the various deformation phases is somehow confusing. Several issues:*

*1) the use of grey literature, such as the Nussbaum unpublished PhD thesis;*

- **Author's reply:** Thank you for pointing out, that using the PhD thesis of Nussbaum is maybe not ideal when it comes to overall deformation phases of the eastern Southern Alps. We agree and therefore removed the reference from the figure caption of Fig. 13.

*2) the occurrence of the pre-Adamello phase in the eastern Alps: why adding this as D2 phase, if the authors (correctly) claim this phase was never recorded in the eastern Southern Alps?*

- **Author's reply:** Thank you for pointing out, that this statement from ours is indeed confusing. Probably biased by unpublished thermochronological data of ours, we also intended to include the pre-Adamello phase in the summary of deformation phases in Fig. 13. We agree, that this is confusing and not related to the data we show in this study and therefore decided to remove the D2 phase from Fig. 13 and from the text in the first paragraph of section 6.2.

*3) The authors propose Miocene (D3), a Miocene to Pliocene (D4) top S, and a final top to the E Plio-Pleistocene phase (D5). However, Castellarin and Cantelli (2000) and Caputo et al. (2010) document variable shortening axis between N340° and N310°, from Serravallian to the Pleistocene. Therefore, how did the authors separate the 3 last phases? Please, explain the method used, or correctly refer the previous authors findings. Finally, in Fig. 14 a2, the top to the S could be due to the mixing of SW and SSE directed faults, maybe formed at different time. This is a common problem of structural sites in the polydeformed Southern Alps, when cross-cutting relationships cannot be found. The authors should consider this alternative interpretation.*

- **Author's reply:** Thank you for making us aware of not correctly referring to previous studies when it comes to the deformation phases from Miocene to Pleistocene. Concerning our own data, we could clearly see cross-cutting criteria in the field of top E directed slip cross-cutting top S to top SSE directed slip. Therefore, we placed the top E directed structures as youngest deformation phase, specifically to Plio- to Pleistocene in agreement with previous studies (e.g., Castellarin and Cantelli, 2000). As you correctly state, cross-cutting relationships can not be found everywhere in the Southern Alps. Exactly this is the problem when it comes to separating top SSW, top S and top SSE directions (our initial deformation phases D3 and D4). Therefore, we decided to combine phases D3 and D4 in Fig. 13 to one phase, which we label D3 top (S)SE in the new version of the manuscript. Please see the modified Fig. 13 and the modification in the text in section 6.2. Additionally, we added a contour plot to support the top S directed fault slip data of Fig. 14 a2. Of course, we can not totally exclude mixing of directions. Therefore, we are grateful for your suggestion to incorporate this topic in our discussion. We therefore modified the text of section 6.2.

*Specific points*

*Line 53 add Bernoulli and Jenkyns, 1974 in the references as the first modern paper dealing with platforms and basins in the Southern Alps.*

- **Author's reply:** Thank you for the suggestion. The reference was added.

*Line 94 Early Permian (geochronology) instead of Lower Permian (chronostratigraphy).*

- **Author's reply:** Modified.

*Line 94 to 97 Actually, the Early Permian event is highly debated but very likely not associated to the rifting of the Hallstatt-Meliata ocean. See f.i. Muttoni et al. 2003 for interpretation associated to large scale intracontinental transform. Most scholars place the start of the first rifting phase in the Late Permian (e.g. Bertotti et al., 1993), ending in the Carnian.*

- **Author's reply:** Thank you for pointing this debate out. We agree on including this debate in our manuscript and therefore modified the text of the first paragraph of section 2.1.

*Line 100, 134 and 722: too many brackets after Vrabeč, Zampieri, Beccaluva and Doglioni.*

- **Author's reply:** Modified.

*103-105 the quoted papers of Martinelli, Pieri & Groppi and Masetti are not suitable for the depth of faulting, since they are not based on observations, purely speculative. Two case studies of depth prolongation of the normal faults in the western Southern Alps are the Lugano fault (e.g. Bertotti, 1990) and the Pogallo fault (Handy, 1987). I would find more convincing quoting these last authors, although they are dealing with structures that slipped more than the ones of the eastern Southern Alps.*

- **Author's reply:** Thank you for your suggestions regarding the references for the depth of faulting. We agree, that Martinelli, Pieri&Groppi, and Masetti only show schematic

figures with schematic depths of normal faults. Therefore, we modified the references in section 2.1 according to your suggestions.

*Line 136 No reference to age in Fantoni and Franciosi (2010) and Vignaroli et al. (2020)! I double checked them, since Late Oligocene SSE directed shortening is in contrast with the tectonic stratigraphy presented by Castellarin et al. (1992), Castellarin and Cantelli (2000). These latter papers, based on data collected on large parts of the Southern Alps, document a Chattian to Burdigalian SSW directed shortening on a regional scale. See General Comments.*

- **Author's reply:** Thank you for pointing this discrepancy out. We removed the references Fantoni and Franciosi (2010) and Vignaroli (2020) and reordered the text in section 2.2.

*Line 150-151 better quote Venzo 1977 for the folded Pliocene.*

- **Author's reply:** Thank you for your suggestion. We modified the reference for the folded Pliocene to Venzo (1977).

*172 Variscan*

- **Author's reply:** Modified.

*175 Late Permian*

- **Author's reply:** Modified.

*178 Please revise this sentence. Besides the two "are", the message is not clear.*

- **Author's reply:** Thank you for pointing this out. The sentence has been revised.

*185 "bioclastic to marly sediments" is not correct for describing the Paleogene to Miocene stratigraphy. Please, use "limestone and marls".*

- **Author's reply:** Modified.

*186 During the Early Jurassic. "Footwall of rifted margins" is not correct: footwall refers to a fault limb; rifted margins refer to a larger scale epicontinental sea. You can use: "footwall of the major normal faults", instead.*

- **Author's reply:** Modified.

*193 evaporite-bearing shales, instead of facies associations.*

- **Author's reply:** Modified.

*340 which simulates*

- **Author's reply:** Modified.

*397 preferentially instead of preferred? Change e.g. (exempli gratia), used when you choose some examples from a larger list, with i.e. (id est), used when, as in this case, the authors deepen the meaning of the previous words.*

- **Author's reply:** Modified.

*445 Fig. 9 appears prior to Fig. 8, which is named first in Line 461. Consider revising the order and number of the Figures…*

- **Author's reply:** This was a mistake in the initial version of the manuscript as it should have been Fig. 7 instead of Fig. 9. Therefore, thank you for pointing this mistake out. The number of the Figure has been changed from Fig. 9 to Fig. 7 in the revised manuscript.

*453 and 471 Compared to, instead of Comparable*

- **Author's reply:** Modified.

*500 Both model 7 and model 8 show. No commas*

- **Author's reply:** Modified.

*610 not clear to what are the authors referring with "former"*

- **Author's reply:** Point taken. We modified the sentence.

*611 …developed, as…*

- **Author's reply:** Modified.

*676 to 678 Unclear this change of scale and the relevance of the Calignano et al 2017 paper. Please, motivate it.*

- **Author's reply:** Thank you for pointing out that mentioning Calignano et al 2017 in this context is not clear. Therefore, we decided to remove this sentence and stick to crustal-scale.

*688 …2022), we focus…*

- **Author's reply:** Modified.

*747 "steep to the S dipping backthrust" Please, change in: backthrust steeply dipping to the S.*

- **Author's reply:** Modified.

*771 …along major fault zones.*

- **Author's reply:** Modified.

*779 D'Alberto*

- **Author's reply:** Modified.

*936 Carnico-Friulano, 1992. Studi Geologici Camerti. Nuova Serie (1992): 275-284.*

- **Author's reply:** Modified.

*1091. Memorie di Scienze Geologiche, Padova, …,*

- **Author's reply:** Modified.

*References*

*Angelier, J. (1979). Determination of the mean principal directions of stresses for a given fault population. Tectonophysics, 56(3-4), T17-T26.*

*Bergerat, F. (1987). Stress fields in the European platform at the time of Africa-Eurasia collision. Tectonics, 6(2), 99-132.*

*Bernoulli, D., & Jenkyns, H. C. (1974). Alpine Mediterranean and central Atlantic Mesozoic facies in relation to the early evolution of the Tethys.*

*Bertotti, G. (1990). Early Mesozoic extension and Alpine tectonics in the western Southern Alps: The geology of the area between Lugano and Menaggio (Lombardy, Northern Italy) (Doctoral dissertation, ETH Zurich).*

*Castellarin, A., Cantelli, L., Fesce, A. M., Mercier, J. L., Picotti, V., Pini, G. A., Prosser, G. & Selli, L. (1992). Alpine compressional tectonics in the Southern Alps. Relationships with the N-Apennines. In Annales tectonicae (Vol. 6, No. 1, pp. 62-94).*

*Castellarin, A., & Cantelli, L. (2000). Neo-Alpine evolution of the southern Eastern Alps. Journal of Geodynamics, 30(1-2), 251-274.*

*De Vicente, G., Vegas, R., Muñoz-Martín, A., Van Wees, J. D., Casas-Sáinz, A., Sopeña, A., ... & Fernández-Lozano, J. (2009). Oblique strain partitioning and transpression on an inverted rift: The Castilian Branch of the Iberian Chain. Tectonophysics, 470(3-4), 224-242.*

*Handy, M. R. (1987). The structure, age and kinematics of the Pogallo Fault Zone; Southern Alps, northwestern Italy. Eclogae Geologicae Helvetiae, 80(3), 593-632.*

*Hippolyte, J.-C., and P. Mann, 2021, Neogene paleostress and structural evolution of Trinidad: Rotation, strain partitioning, and strike-slip reactivation of an obliquely colliding thrust belt, in C. Bartolini, ed., South America–Caribbean–Central Atlantic plate boundary: Tectonic evolution, basin architecture, and petroleum systems: Tectonic evolution, basin architecture, and petroleum systems: AAPG Memoir 123, p. 317–346. DOI: 10.1306/13692249M1233851*

*Muttoni, G., Kent, D. V., Garzanti, E., Brack, P., Abrahamsen, N., & Gaetani, M. (2003). Early Permian Pangea 'B' to Late Permian Pangea 'A'. Earth and Planetary Science Letters, 215(3-4), 379-394.*

*Simón, J. L. (2019). Forty years of paleostress analysis: has it attained maturity? Journal of Structural Geology, 125, 124-133.*

*Varga, R. J. (1993). Rocky Mountain foreland uplifts: Products of a rotating stress field or strain partitioning? Geology, 21(12), 1115-1118.*

*Venzo, S., with collaboration of Petrucci, F., & Carraro, F. (1977). I depositi quaternari e del Neogene Superiore nella bassa valle del Piave da Quero al Montello e del Paleopiave nella valle del Soligo (Treviso). Memorie degli Istituti di Geologia e Mineralogia dell'Universita di Padova, 30, 1–27.*

**2. Point-by-point response to reviewer 2, Pamela Jara**

Dear Pamela Jara,

We are grateful for your valuable comments and suggestions which particularly helped improve on clarity and readability of the manuscript. All specified points have been considered, text and figures have been modified based on the comments, which we refer to in our co-listing reply below.

Please do not hesitate to reach out to us for further clarification in any of our responses to your review.

Best,
Anna-Katharina Sieberer and co-authors.

*General comments*

*This contribution attempts to explain a complex Alpine deformation zone through analogue modeling, which is a tool that allows simplifying the processes in order to understand the variables that affect the particular study case. It is an interesting study, and it is undoubtedly a contribution since continental polyphasic deformation is something that is observed in various parts of the Earth, so the variables under analysis can be applied to other case studies. The study is very complete in terms of its geological setting and deformation phases involved, so it has strong support prior to proposing the models to be developed in the following sections. However, from section 3 (analogue modeling approach) it becomes confusing, because 12 models are developed, but they are not properly grouped to be able to make an ordered comparison of the variables included in each of the 12 cases. On this, some more detailed comments are made in the following section.*

*As a general comment, there seem to be 2 or 3 sets of models (which include 2 or 3 models each set) comparable to each other, there are even two large sets of models (brittle only and brittle/ductile) that it is not clear that they can be comparable, since they have different scaling rati.*

*It is proposed that the authors reorganize their set of models in order to clarify which models are comparable to each other, and which factor is comparable (which is the variable) in each case of comparison, since the 12 models are different in different variables, but they can be grouped in pairs to compare. The proposal is to generate clear codes for each model and build a comparative table both initially (to present the models and their variables) and final (to present the results of the comparisons). Ex: models 6 and 8 are both brittle/ductile, the difference between is the angle of obliquity for inversion (0 and 20) and that is the variable to be evaluated when it is performed this comparison; so they can make a table in which that is highlighted (and models could be renamed as BD_0 and BD_20).*

- **Author's reply:** Thank you for this comment, which makes clear that our original grouping of experiments as presented in section 4.2 lacks clarity. Following the reviewer's suggestion, we modified Table 1, which now provides detailed information on how the models are grouped and what their underlaying variables are. Comparison of modelling results is indeed most appropriate for models within a certain group. However, we remark that the length-scale ratio, which controls the peak strength at the

base of the brittle crust, is the same in all models. We argue that a comparison of modelling results across the defined groups is valid. For further enhancing clarity on the tested parameter space, we adapted our strength profiles in Fig. 4 by explicitly accounting for friction and cohesion of the brittle materials, following Weijermaars (1997, Principles of Rock Mechanics).

We also reordered some models and consecutively numbered all experiments in the order of their description from 1-12. As a consequence, we also had to reorganise Figs. 8, 9, 10, 11, 12, and 14 and adapted, rearranged and simplified some text in section 4.2 to warrant coherence and readability of the manuscript. We remark that Fig. 11 provides in visual attractive way an overview of modelling results, which facilitates picking up the main differences related to the tested parameter space. We prefer this way of presentation than adding another table.

The suggestion of coding based on model characteristics may well work for simple experiments. As our polyphase models are quite complicated, this would lead to long and complicated codes. As such we prefer to stick to our simple numeric labelling (model 1 - model 12). Finally, we included a short introduction to the 3 sets of analogue models in the text of section 4.2 to better define the tested parameter space per experimental set.

*Later on, we talk about a "reference model", but the reference conditions are not specified, nor is highlighted the variable to be compared with the other 11 models. Better explain the reference model and against which models it is correct to use it as a reference. I think that in this case it is better to separate the 12 models into 2 or 3 large sets and compare them properly with each other, since there are many variables at stake.*

- **Author's reply:** Point taken. Our reference model is the most basic of all models, which allows for comparing it with the other experiments as only one parameter as been changed per model with respect to the reference model. We added a few lines in section 4.1 explaining better the conditions and role of the reference model in this study.

*I believe that by reorganizing the presentation of the models and variables under study and comparison, a good part of the following is clarified, and the work will look much better presented. Thus, it is a great contribution to the understanding of variables that affect complex polyphase deformation systems where many variables contribute.*

- **Author's reply:** We agree, and followed the reviewer's suggestions (see detailed response above), which indeed lead to a better and clearer presentation of our analogue experiments (see also Table 1).

*Detailed suggestions*

*Line 199: I suggest listing here (this introductory paragraph to the methodology) the similarities and differences between the 12 models in order to identify the variables in each case.*

- **Author's reply:** We agree with the reviewer that introducing the tested variables at this stage of the manuscript is important. We, therefore, rephrased our previous wording on this to specify the parameter space, which results in grouping of the analogue experiments. In our view, a detailed description of similarities and differences among

the 12 presented models is better placed in section 4.2 (parametrical study) as the introduction to the analogue modelling sections provides the general modelling approach whereas details of the models are explained in section 4.2, where we added more detailed descriptions of the sets of experiments in sections 4.2.1, 4.2.2 and 4.2.3.

*Line 212: (VD) is not clear in the mentioned figure 3a-c: To clarify, is it referring to VD1 and VD2?, since in c they do not coincide. I suggest modifying Fig.3 in order to include an original set-up (pre-extension), indicate the meaning of the arrows and add some scaling details.*

- **Author's reply:** Point taken. We accordingly modified "(VD)" to "(VD1 and VD2)" (section 3.1), explained their respective meanings (section 3.1), and added a pre-extension sketch of our modelling setup in Fig. 3. Furthermore, we made differences in kinematics of the deformation phases explicit (grey and black arrows) and added amounts of displacement for both extensional and compressional phases. Other scaling details (e.g., velocities of pulling and pushing for various materials) are provided in Table 1.

*Line 212: ..."and thus the strike of the basin axes"..: this seems to be a hypothesis or a result. Clarify, since the direction of stress in extension (arrows in fig 3) is not aligned.*

- **Author's reply:** Point taken. The direction of arrows in Fig. 3 indicates the direction of the mobile plates, rather than the extension direction at the rifts, which is controlled by the orientation of the velocity discontinuities (VD1 and VD2). We clarified this item in the caption of Fig. 3 where we explain the meaning of the grey and black arrows, which relate to the motion of the mobile plates.

*Line 213: "Pre-deformation rotation…": shortening?*

- **Author's reply:** Point taken. We rephrased and simplified the sentence to: "Positioning the VDs obliquely allows for the formation of graben structures, which are at angles of 10 and 20 degrees with respect to the shortening direction (Fig. 3a-c)."

*Line220: Are these values for all experiments? if so, add it in Fig.3 in the general set-up.*

- **Author's reply:** Yes, those values are used in all experiments as now indicated in Fig. 3a-c.

*Line 221: "...producing sedimentary basins of different size…" : seems to be a result, it should be mentioned after the development of the experiences.*

- **Author's reply:** "Producing sedimentary basins of different sizes" is indeed a model result related to the first phase of deformation. However, as this result is critical for making the bridge to the natural example and for understanding the subsequent inversion related deformation geometries, we prefer to keep this sentence in section 3.1.

*Line 235 to 239: This paragraph describes some characteristics of the different 12 models, however models 9 and 10 are not mentioned.*

- **Author's reply:** Thank you for pointing this out. We included the missing models in the description (section 3.1).

*In general, the description of the models is very confusing, since they do not have any order or assigned codes that allow them to be associated with the descriptions. It is proposed to generate some coding, and a table of factors (variables) to be analyzed for each model.*

- **Author's reply:** In the original manuscript, the grouping of experiments has been done on the basis of the explored parameter space (sections 4.2.1 – 4.2.3), which however seemed to have served its purpose insufficiently. In the revised manuscript we modified Table 1 to foster clarity on the tested parameters and the grouping of the experiments. However, as explained earlier, we prefer to stick to a simple numeric coding (model 1 – model 12) of the experiments.

*Figure 4: a) MOD10 the symbology is not included in the legend of the figure. B) MOD6 and MOD8 (brittle/ductile) cannot be compared to filled basin (is that correct?).*

- **Author's reply:** a) Thank you for pointing this out. We included the symbology for model 12 (model label after reorganisation, check Table 1) in the legend of Fig. 4. b) Yes, that is correct, as for models with ductile décollement only underfilled basins were modelled (models 6 and 7 in the new version of the manuscript).

*Line 242: Explain why the difference between the speeds when extending and compressing, and between brittle/ductile or brittle-only models (EXT: 5 or 2.5 cm/h and COMP: 3 or 2.5 cm/h).*

- **Author's reply:** The reason for modelling with different velocities for brittle only and brittle/ductile experiments is the time dependence of ductile materials. In order to obtain strength values for ductile layers, which allow for decoupling of the ductile layer (a condition needed for acting as a décollement), an adjustment of the velocity was needed. We chose a faster velocity for the extension phase in purely brittle models for reasons of convenience. We added the additional description in the text (section 3.1).

*Line 249: Table1: I suggest reorganizing with codes and order that denote some characteristic of the model, and highlight the variable that allows making the respective comparisons (Ex: B_0 vs B_10 both brittle only but with differences in…).*

- **Author's reply:** Point taken. Please see our responses above to the same comment.

*Line 257: …"whereas the ductile layer consists of polydimeth…": Indicate if it represents something in the real study case (o detachment layer or similar) or if it is just a device that allows you to geometrically modify where to generate the deformation.*

- **Author's reply:** We stress that arguments for using a ductile layer have been put forward in the original manuscript (line 270). We now return to this item and included the natural pendant for the use of ductile material in our analogue experiments, as simulation for the ductile middle crust, in the text of section 3.2.

*Line 275: Ductile layer in MOD 6 and 8 (brittle/ductile) has the same depth as the detachment of MOD 5-7 and 10 (brittle only), but different scale ratio, is that correct?. In addition, the thickness in both cases is the same (silicone layer and layer of glass beads in Fig. 4). Explain this scaling.*

- **Author's reply:** Yes, the ductile detachment of models 6 and 7 (model labels after reorganisation, check Table 1) is at the same depths and of the same thickness as the frictional detachment (glass beads in models 4, 5, and 12). For those models just

mentioned, the scaling is the same for the quartz sand (brittle layer above the ductile detachment or the frictional detachment) but different for the detachment layer (in models 6 and 7 ductile detachment vs. in models 4, 5, and 12 frictional detachment). In order to emphasise these differences, we adapted and reorganised the strength profiles in Fig. 4 and added the frictional décollement to clarify and make the comparison easier. Overall, the length-scaling is the same in all experiments.

*Line 290: add to table 2 (which only refers to brittle materials) the properties of the other materials used in the modeling and which are mentioned in this paragraph.*

- **Author's reply:** Thank you for this valuable suggestion. We added the parameters for the ductile material to Table 2.

*Line 320: the "reference model" : Does it serve as a reference for the other 11 models in this study? To justify.*

- **Author's reply:** We clarified the role of the reference model in section 4.1 (see also earlier responses to the same comment).

*Until this line (320) the difference, for example between models 1, 2 and 3, is not yet known, since in Fig. 3 all 3 are presented in the same conditions. It is necessary to create a table (and figure) where the conditions of the "reference model" are highlighted. If this model is compared with the other 11, then also highlight which is the variable for each case; otherwise "pair" according to whether the comparisons are made in another way in the following sections.*

- **Author's reply:** Thank you for your suggestions concerning our reference model. As explained earlier we modified Table1 as suggested. We also changed the position of Table 1 within the manuscript (now embedded at the end of the introduction to section 3) in order to define the sets of experiments earlier, which benefits the clarity of the manuscript.

*It is not clear why the following figures compare:*

*-only models 2 and 3 (Fig. 6 and 7), then models 7 and 8 (Fig. 8 and 9). Do we skip models 4 and 5?*

- **Author's reply:** Thank you for pointing out the inconsistency in numbering of the experiments. We did not skip models 4 and 5, but they have been presented at a different place. After re-naming, models 4 and 5 became models 8 and 4, respectively. Using the new labelling, model 8 is compared with models 9 and 10 in Fig. 10. We agree, that a better ordering of our experiments will lead to a better understanding of what compares to what, hence the modified Table 1. Also, we need to mention, as we do not have the possibility according to the length of the manuscript, to show all models in detailed comparison, we decided to only compare 2 out of the 4 experiments in set 2 in detail. We explain in sections 4.2.2 and 4.2.3, why we do not provide detailed explanations of models 4 and 6 (section 4.2.2) and of models 11 and 12 (section 4.2.3).

*-models 4, 11 and 12 (Fig. 10) . Why was it decided to number that way if there are intermediate models not discussed? is confusing.*

- **Author's reply:** Thank you for making us aware of better structuring our experiments. We agree that our numbering is confusing and therefore reordered the experiments (see Table 1 and earlier responses).

*- Figures 11 and 12 show plans and graphs of the 12 models. It is important to know if the 12 are comparable to each other indistinctly, otherwise it might be better to group them (pair them for example) for a better comparison considering the variables at stake (may be in the same figure but in an order that allow comparison).*

- **Author's reply:** Point taken. We agree that our numbering is confusing and reordered the experiments in Figs. 11 and 12 based on our improved model grouping and sequence, presented in Table 1.

**3. Further changes in the manuscript, not related to comments of reviewer 1 or 2**

During the review of our manuscript, we figured out another way to describe our major results, the lateral variations of thrust orientations across platform-basin boundaries. Therefore, in the revised manuscript, we describe those changes in thrust orientation with respect to the overall strike of the orogen and not to the shortening direction (as described in the original manuscript). This required some overall rewording throughout the revised manuscript, In detail, for our reference model (model 1) we, e.g., added the reworded description in lines 411-414 (see track change file of the revised manuscript version). For models of our experimental set 2, we changed the description between lines 448-457 (see track change file of the revised manuscript version), for models of set 3 between lines 545-548.

---

## Author Response (AR2)

Dear Riccardo Reitano,

We are grateful for your comment which helped improve on clarity of the manuscript. The specified point has been considered and the sentence rephrased, which we refer to in our co-listing reply below.

Best,
Anna-Katharina Sieberer and co-authors.

***Public justification (visible to the public if the article is accepted and published)***:

*Lines 372-373: I would rephrase the sentence explaining better the role of reference model. It is the simplest model in the way that is the models used as "starting point" from which the Authors changed parameters, adding more complexities that control models' evolution. The effect of the parametric study on models' evolution is explainable because there is a reference to compare results to, as the Authors did in section 4.2.*

- **Author's reply:** Thank you for your comment on adding further explanation on the role of the reference model. We very much agree with your suggestion and modified the sentence according to your suggestion to: "Model 1 serves as reference model as it is the relatively simplest experiment of our study, representing the starting point for changes of various parameters (parametric study, section 4.2) and for the gradual addition of complexity in the subsequent models."